# Unified Theory of Adaptive Variance Reduction

## Abstract

Variance reduction is a family of powerful mechanisms for stochastic optimization that appears to be helpful in many machine learning tasks. It is based on estimating the exact gradient with some recursive sequences. Previously, many papers demonstrated that methods with unbiased variance-reduction estimators can be described in a single framework. We generalize this approach and show that the unbiasedness assumption is excessive; hence, we include biased estimators in this analysis. But the main contribution of our work is the proposition of new variance reduction methods with adaptive step sizes that are adjusted throughout the algorithm iterations and, moreover, do not need hyperparameter tuning. Our analysis covers finite-sum problems, distributed optimization, and coordinate methods. Numerical experiments in various tasks validate the effectiveness of our methods.

## 1 Introduction

In this paper, we are interested in the optimization problem

$$\min_{x \in \mathbb{R}^d} f(x). \tag{1}$$

This setting is used in various fields, such as engineering (Snyman et al., 2005), statistics (Shalev-Shwartz & Ben-David, 2014), machine learning (Goodfellow et al., 2016), etc. There are many approaches to solve (1), and gradient methods are among the most established ones (Ruder, 2016; Haji & Abdulazeez, 2021).

With the increasing complexity of datasets and the expanding parameters of models (Naveed et al., 2023), numerous heuristics has been adopted within the learning process to boost its efficiency. Many of them employ stochastic gradient estimations that greatly minimize the cost of each iteration without hindering the convergence process. Apart from the standard vanilla SGD (Robbins & Monro, 1951; Moulines & Bach, 2011), multiple approaches have been introduced to reduce the difference between the actual gradient and its estimator.

This strategy utilizing inexact gradients has shown success in various contexts, such as finite sum problems often encountered in machine learning and distributed optimization needing good communication. These techniques are designed to capture the most critical information from the minimized function, thereby preserving the convergence characteristics while reducing computational costs. From the theoretical point of view, tuning of all these methods depends either on the problem's smoothness constant or on the gradients' upper bound, which might not be known beforehand. To address these challenges, numerous SGD-like adaptive techniques have been introduced (Zhou et al., 2018), focusing on utilizing information from present and prior iterations to approximate the problem's parameters and define upcoming step sizes. Though this problem has been familiar for a long time (Polyak, 1987), recently it has been revisited numerous times, for instance in AdaGrad (Duchi et al., 2011), Adam (Kingma, 2014), Prodigy (Mishchenko & Defazio, 2023), and others. However, the main spotlight in these papers was on SGD, rather than variance reduction methods.

In this paper, we connect these two approaches of the stochastic optimization: adaptivity and variance reduction, and develop new schemes, that benefit from all of the concepts mentioned above.

## 2 RELATED WORK

### MANY FACES OF (STOCHASTIC) GRADIENT

The SGD update scheme is simple and can be generalized as below:

---
**Algorithm 1** General Scheme of SGD

---
**for** $t \in 1 \ldots T - 1$ **do**
  Compute step size $\gamma_t$
  Generate stochastic $\xi_t$
  Compute estimator of $\nabla f(x^t) : g^t = g^t(x^t, \xi_t, history)$
  Update $x^{t+1} = x^t - \gamma_t g^t$
**end for**

---

Over the recent years many techniques has been developed, which aim to deal with non-vanishing variance of SGD. Starting with the finite sum problem:

$$\min_{x \in \mathbb{R}^d} \left\{ f(x) = \frac{1}{n} \sum_{i=1}^{n} f_i(x) \right\}, \tag{2}$$

such estimators as SAG (Roux et al., 2012; Schmidt et al., 2017), SAGA (Defazio et al., 2014), SVRG (Johnson & Zhang, 2013), SARAH (Nguyen et al., 2017), PAGE (Li et al., 2021a) and many others were proposed. These methods exploit the structure of $f$ and use different strategies to learn the gradient recursively by stochastic sampling on every iteration. This leads to the noise decrease as converging to the optimum, which is not obtained in the SGD framework.

While in finite sum setting $f_i$ in the equation (2) stand for the loss on distinct samples, in the distributed setting, these $f_i$ are stored on the different nodes and represent the losses, computed on local datasets. In this case, $n$ stands for the number of nodes. We consider the setup, where all nodes communicate with the server, that aggregates the information and transfer the new state to the devices. Frequently, the local gradients are transmitted from the nodes to the central server. In contrast to the local scenario, the main obstacle in this case is the communication bottleneck – we need to obtain the optimum with less number of bits sent. To mitigate the transmission costs, various compression mechanisms are incorporated, such as quantization (Gupta et al., 2015; Beznosikov et al., 2023b) and sparsification (Alistarh et al., 2018). They are utilized in advance distributed optimization methods, like DIANA and MARINA. (Mishchenko et al., 2019; Gorbunov et al., 2021), inspired by variance reduction technique. Later, another scheme with the error compensating technique (Richtárik et al., 2021) appeared, where a broader class of biased compressors can be utilized instead of unbiased ones.

Another illustrations of stochastic optimization are randomized coordinate methods, for instance, SEGA (Hanzely et al., 2018) and JAGUAR (Veprikov et al., 2024). It appears, that these algorithms also can be viewed as variance reduction, since the difference between the exact gradient and its estimation can be bounded recursively, which is the main property of the methods above.

In the recent years, many unified analysis for stochastic first-order methods under various assumptions were developed, which aim to unite diverse gradient-based methods under one umbrella. They covered many cases, however, still there are some gaps in theory. One of the first analyses (Gorbunov et al., 2020; Li & Richtárik, 2020) demanded an unbiased estimation ($\mathbb{E}g^t = \nabla f(x^t)$). However, many other methods were not covered. For instance, in (Driggs et al., 2022) the authors required the gradient estimators to be the memory-biased or recursively biased $\left( \mathbb{E}[g^t - \nabla f(x^t)] = (1 - \rho)[g^{t-1} - \nabla f(x^{t-1})] \right)$. But there analysis was applicable only to the small number of algorithms, such as SAGA, SARAH and SVRG. Overall, no comprehensive analysis exists, that include various setups and different gradient estimators.

### ADAPTIVE LEARNING RATE

Instead of using the constant step sizes, that are predetermined, many algorithms, as deterministic as well as stochastic, are designed to adjust the learning rate throughout the iteration process. This allows to accelerate the convergence in the beginning, where we are far away from the optimum, and to take more precise steps when we are near the solution.

This setup is not new - selecting learning rates, based on the method behaviour on particular problem has been analyzed in the previous century. Armijo (Armijo, 1966) and Wolfe (Wolfe, 1969) rules are used to select the step size with demanded decrease. Nesterov (Nesterov, 1983) proposed a backtracking method for finding the local smoothness constant, that is updated at each iteration. However, this approach is resource-consuming, as it requires multiple gradient evaluations. Furthermore, some schemes are applicable only to convex functions. Polyak (Polyak, 1987) proposed a step size, that utilized the relative functional suboptimality, as well as the gradient's norm. This approach was recently revived in machine learning, and investigated by several works (Hazan & Kakade, 2019; Takezawa et al., 2024). The main weakness of these methods is the dependence of minimum function value, which might not be known beforehand.

Another approach is aimed to function or gradient's Lipschitz constant. Such methods, as AdaGrad (Duchi et al., 2011), RMSprop (Tieleman, 2012), Adam (Kingma, 2014), AdamW (Loshchilov & Hutter, 2017) etc. All these optimizers demonstrate a decent performance on various machine learning problems, however, they lack of theoretical justification and also require hyperparameter tuning.

Inspired by AdaGrad technique, variance reduction method STORM (Cutkosky & Orabona, 2019) was developed, which provably improves the bounds of SGD. It combined SAGA and SARAH with adaptive step sizes and achieves better convergence, than algorithms with constant learning rate. This method was followed by STORM+ (Levy et al., 2021), Ada-STORM (Weng et al., 2017), SAG-type STORM (Jiang et al., 2024). The problem, still, is in tuning the hyperparameters.

There exist parameter-free methods, that are thoroughly designed to adjust step size without tuning. For instance, Bisection (Carmon & Hinder, 2022), that iteratively approximate smoothness of the initial problem, D-Adaptation (Defazio & Mishchenko, 2023) and Prodigy (Mishchenko & Defazio, 2023), which are AdaGrad variations with additional estimating the distance towards the solution. Also, other approaches, based on online optimization (McMahan & Streeter, 2010), exist. These methods aim to estimate the unknown parameters of the problem via the known statistics, such as gradient norms. Another advantage is the ability to deploy the learning process without adjusting a big number of hyperparameters - this is especially valuable in large models, which training must be resource efficient.

While all adaptive and parameter-free methods can be regarded as variations of SGD, no extension for distributed and coordinate methods were analyzed. Furthermore, only a small number of variance reduction algorithms were combined with these approaches, often demanding a varying set of assumptions and not always providing optimal convergence rates.

## 3    OUR CONTRIBUTION

• **New adaptive methods.** We suggest a wide family of stochastic methods that are implemented with adaptive step sizes and do not depend on the smoothness constant. It is worth noting, that asymptotically these rates matches with the best known for these methods.
• **Unified scheme.** We propose the new unified analysis for variance reduction modifications of stochastic gradient descent. It does not require the unbiased gradient estimators, which allows to include more method than previous analyses.
• **Experiments.** We show through rigorous experiments that proposed methods show compatible performance with the existing ones. Experiments for stochastic, coordinate and distributed methods are provided.

## 4    MAIN PART

In this section, we introduce all the necessary assumptions and elaborate on the methods and convergence rates.

**Notation.** We use the standard Euclidean norm for vectors: $\|x\| \stackrel{\text{def}}{=} \langle x, x \rangle^{1/2}, x \in \mathbb{R}^d$. The objective functional $f : \mathbb{R}^d \to \mathbb{R}$ is a differentiable function. We denote its global minimum by $f_* \stackrel{\text{def}}{=} \inf_{x \in \mathbb{R}^d} f(x) > -\infty$ which may not be unique. We also introduce the gradient of $f$ at point $x$ as $\nabla f(x) \in \mathbb{R}^d$.

**Definition 1.** *Function $f$ is called $L$-smooth, if there exists $L \geq 0$ such that*
$$\|\nabla f(x) - \nabla f(y)\| \leq L\|x - y\| \quad \forall x, y \in \mathbb{R}^d.$$

**Definition 2.** *Function $f$ satisfies Polyak-Lojasiewic (PL) condition, if there exist $\mu > 0$ such that*

$$f(x) - f_* \le \frac{1}{2\mu} \left\| \nabla f(x) \right\|^2 \quad \forall x \in \mathbb{R}^d.$$

Smoothness condition is standard in stochastic optimization. PL condition is also frequently met in theory, since over-parameterized neural networks are locally PL (Liu et al., 2022).

UNIFIED ASSUMPTION

The next assumption is the key one in this manuscript, as it describes the behaviour of the stochastic gradient estimation. If the iterations are conducted according to Algorithm 1, then the behaviour of the convergence process fully depends on the choice of $\{\gamma_t\}$ and $\{g^t\}$. To describe the recursive nature of variance reduction we introduce the following:

**Assumption 1.** *Let $\{x^t\}$ be the iterates of Algorithm 1 and $\{\xi_t\}$ - random variables, generated by it. Define $\mathcal{F}_t = \sigma(x_0, \ldots, x_t, \xi_1, \ldots, \xi_{t-1})$. Let there be non-negative constants $A, B, C$ and $\rho_1, \rho_2 \in (0, 1]$ and a (possibly) random sequence $\{\sigma_t^2\}$, such that for $\forall t$ the following inequalities hold:*

$$\mathbb{E}\left[ \left\| g^t - \nabla f(x^t) \right\|^2 \mid \mathcal{F}_t \right] \le (1 - \rho_1) \left\| g^{t-1} - \nabla f(x^{t-1}) \right\|^2 + A\sigma_{t-1}^2 + BL^2 \|x^t - x^{t-1}\|^2,$$

$$\mathbb{E}\left[ \sigma_t^2 \mid \mathcal{F}_t \right] \le (1 - \rho_2)\sigma_{t-1}^2 + CL^2 \|x^t - x^{t-1}\|^2. \tag{3}$$

Let us discuss the meaning of the constants in the equations above. We demand the proposed methods in a sort of way to be not expanding, this guarantees, that the differences between the estimator and the exact gradient mitigate as the optimum is approached. This is assured by constants $\rho_1$ and $\rho_2$, since they are strictly more than zero. Parameter $A$ is need for the same purposes - it connects the difference between the error is estimation with additional sequence. As most of considered methods utilize estimators, that incorporate the gradient information from previous steps, constants $B$ and $C$ are used to bound the difference with the step size. This implies, that as steps diminish near the extremum point, the estimators are more precise.

Especially, it should be noted, that constants $\rho_1, \rho_2, A, B, C$ depend entirely on the estimator properties, i.e., number of devices $n$, batch sizes $b$, probability $p$, compressor's qualities, dimensionality $d$, etc. And they are independent of any information, depending on data, for instance smoothness constant L and PL constant $\mu$. Also, these constants are independent of initial or current distance to the solution, functional gap to the optimal value or other information, that encodes the current suboptimality, which is not known beforehand.

We do not demand $g^t$ to be the unbiased estimation of $\nabla f(x^t)$, which is required in (Li & Richtárik, 2020; Gorbunov et al., 2020). This allows to examine a wider class of estimators, than in previous manuscripts. Neither we demand large batches, that mitigate the difference between the gradient and initial approximation (Cutkosky & Mehta, 2021).

Though, additional random sequence was also utilized in previous unified analysis, the unbiasedness allowed to analyze $\mathbb{E}\|g^t\|^2$ instead of $\mathbb{E}\|g^t - \nabla f(x^t)\|^2$. Also, previous papers required $f_*$ in unified assumption, therefore, no recursive contracting nature was captured (Li & Richtárik, 2020). Furthermore, several papers were done in $\mu$-strongly quasi-convex setting, which is restrictive (Gorbunov et al., 2020).

CONVERGENCE GUARANTEES

Now that we have introduced the main assumption, we are ready to derive the theorems, describing the convergence process. To justify the introduced assumptions we start with the non-convex and PL non-adaptive setup, as in other unified analyses.

In the general non-convex setup any method of our scheme converges sublinearly

**Theorem 1.** *Let $f$ be $L$-smooth and satisfy Assumption 1. Then Algorithn 1 with step size*

$$\gamma_t \equiv \gamma \le \frac{1}{L} \left( 1 + \sqrt{\frac{B\rho_2 + AC}{\rho_1 \rho_2}} \right)^{-1},$$

*for any $T > 0$ achieves*

$$\frac{1}{T} \sum_{t=0}^{T-1} \mathbb{E} \|\nabla f(x^t)\|^2 \le \frac{2V^0}{\gamma T},$$

*where $V^0 = f(x^0) - f_* + \frac{\gamma}{2\rho_1} \|g^0 - \nabla f(x^0)\|^2 + \frac{\gamma A}{2\rho_1 \rho_2} \sigma_0^2$.*

After getting to the neighbourhood of the extremum point, we can derive linear convergence of variance reduction methods.

**Theorem 2.** *Let $f$ be $L$-smooth, satisfy PL condition and Assumption 1. Then, Algorithm 1 with step size*

$$\gamma_t \equiv \gamma \le \min \left\{ \frac{1}{L} \left( 1 + \sqrt{\frac{B\rho_2 + 4AC}{\rho_1 \rho_2}} \right)^{-1}, \frac{\min\{\rho_1, \rho_2\}}{2\mu} \right\},$$

*for any $T > 0$ achieves*

$$V^T \le (1 - \gamma \mu)^T V^0,$$

*where $V^t = f(x^t) - f_* + \frac{\gamma}{\rho_1} \|g^t - \nabla f(x^t)\|^2 + \frac{2\gamma A}{\rho_1 \rho_2} \sigma_t^2$.*

The main contribution of this manuscript is variance reduction's compatibility with adaptive methods. Below we define the step sizes, that allow to converge sublinearly.

**Theorem 3.** *Let $f$ be $L$-smooth and satisfy Assumption 1. Then, Algorithm 1 with step sizes*

$$\gamma_t = \frac{1}{\left( \max \left\{ \sqrt{\frac{B\rho_2 + AC}{\rho_1 \rho_2}}; 1 \right\} \right)^{1-\alpha} \left( \sum_{i=0}^{t-1} \|g^i\|^2 \right)^\alpha},$$

*for any $T > 0$ achieves*

$$\frac{1}{T} \sum_{t=0}^{T-1} \mathbb{E} \|\nabla f(x^t)\| \le \mathcal{O} \left( \frac{V_0^{\frac{1}{2(1-\alpha)}} + L^{\frac{1}{2\alpha}}}{\sqrt{T}} \max \left\{ \left( \frac{B\rho_2 + AC}{\rho_1 \rho_2} \right)^{1/4}; 1 \right\} \right),$$

*where $\alpha \in (0, \frac{1}{3})$.*

Note, that in Theorem 3 we bound the average norm if the gradient, while in Theorem 1 the square of the norm.

Since constants $\rho_1, \rho_2, A, B, C$ do not depend on $L, \mu, \|x^0 - x_*\|^2$, where $x_* \in \arg\min_x f(x)$, and so on, steps in Theorem 3 are truly parameter-free, as they are defined only by the estimator's property.

Another question is the choice of the constant $\alpha$. One option is to choose $\alpha = \arg\min V_0^{\frac{1}{1-\alpha}} + L^{\frac{1}{\alpha}}$. However, as these constants might not be known beforehand, more practical option is to choose it, depending on the robustness of method. Experiments show (see Additional Numerical Experiments in Appendix), that smaller $\alpha$ lead to higher variance in the gradient norm, whereas, higher ones result in more robust iterations.

## 5 FAMILY OF METHODS

### 5.1 FINITE SUM PROBLEM

As already mentioned, the problem (2) is frequently met in modern applications, as it can be regarded as empirical risk minimization. Since computing full gradient is expensive, significantly smaller batches can be considered. However, as simple utilizing random batches lead to convergence to some solution's neighbourhood, various gradient approximations are incorporated to boost the performance. Below we examine these schemes.

**L-SVRG.** We consider L-SVRG (Kovalev et al., 2020), the loopless version of SVRG (Johnson & Zhang, 2013). The $g^t$ update can be written in a following way:

$$w^t = \begin{cases} x^{t-1} & \text{with probability } p \\ w^{t-1} & \text{otherwise} \end{cases}, \quad g^t = \frac{1}{b} \sum_{i \in S_t} (\nabla f_i(x^t) - \nabla f_i(w^t)) + \nabla f(w^t), \quad (4)$$

where mini-batches $S_t$ of size $b$ are generated uniformly and independently at each iteration. If the probability is close to one, then $w^t$ is updated quite often and gradient estimation is more based on stochastic mini-batches.

**Lemma 1.** *L-SVRG (4) satisfies Assumption 1 with* $\rho_1 = 1, A = \frac{2}{b}, B = \frac{2}{b}, \sigma_t^2 = \frac{1}{n} \sum_{i=1}^{n} \|\nabla f_i(w^{t+1}) - \nabla f_i(x^t)\|^2, \rho_2 = \frac{p}{2}, C = 1 + \frac{2}{p}$.

**Corollary 1.** *In the non-convex case choosing step sizes as*

$$\gamma \lesssim \left( L \left[ 1 + \frac{1}{p\sqrt{b}} \right] \right)^{-1} \quad \text{results in} \quad \mathbb{E} \|\nabla f(x^\tau)\|^2 = \mathcal{O} \left( \frac{1}{T} \left( 1 + \frac{1}{p\sqrt{b}} \right) \right). \tag{5}$$

*Taking adaptive step sizes as*

$$\gamma_t = \frac{1}{\left( \max \left\{ \frac{1}{p\sqrt{b}}; 1 \right\} \right)^{1-\alpha} \left( \sum_{i=0}^{t-1} \|g^i\|^2 \right)^\alpha} \quad \text{results in} \quad \mathbb{E} \|\nabla f(x^\tau)\| = \mathcal{O} \left( \frac{\max \left\{ \frac{1}{\sqrt{p\sqrt{b}}}; 1 \right\}}{\sqrt{T}} \right). \tag{6}$$

*Here $\tau$ is chosen uniformly over $0, \dots, T-1$.*

To find the optimal batch size $b$ and probability $p$ for the convergence guarantees one should minimize the expected number of gradient calls. This is achieved by analyzing the number of calculated derivatives per iteration, multiplied by the For L-SVRG the expression is $(1 + \frac{1}{pb^{1/2}})(pn + b)$. We obtain $b = n^{2/3}$ and $p = \frac{b}{n} = n^{-1/3}$.

**SAGA.** Another approach is SAGA algorithm (Defazio et al., 2014), where instead of points, stochastic gradients are stored:

$$y_i^t = \begin{cases} \nabla f_i(x^{t-1}) & \text{for } i \in S_t \\ y_i^{t-1} & \text{otherwise} \end{cases}, \quad g^t = \frac{1}{b} \sum_{i \in S_t} (\nabla f_i(x^t) - y_i^t) + \frac{1}{n} \sum_{j=1}^{n} y_j^t, \tag{7}$$

where mini-batches $S_t$ of size $b$ are generated uniformly and independently at each iteration. We collect "delayed" full gradient in $\sum_{j=1}^{n} y_j^t$, which is used to compensate the error in estimation.

**Lemma 2.** *SAGA (7) satisfies Assumption 1 with* $\rho_1 = 1, A = \frac{1}{b} \left( 1 + \frac{b}{2n} \right), B = \frac{2}{b} \left( 1 + \frac{2n}{b} \right), \sigma_t^2 = \frac{1}{n} \sum_{i=1}^{n} \|\nabla f_i(x^t) - y_i^t\|, \rho_2 = \frac{b}{2n}, C = \frac{2n}{b}$.

**Corollary 2.** *In the non-convex case choosing step sizes as*

$$\gamma \lesssim \left( L \left[ 1 + \frac{n}{b^{3/2}} \right] \right)^{-1} \quad \text{results in} \quad \mathbb{E} \|\nabla f(x^\tau)\|^2 = \mathcal{O} \left( \frac{1}{T} \left( 1 + \frac{n}{b^{3/2}} \right) \right). \tag{8}$$

*Taking adaptive step sizes as*

$$\gamma_t = \frac{1}{\left( \max \left\{ \frac{n}{b^{3/2}}; 1 \right\} \right)^{1-\alpha} \left( \sum_{i=0}^{t-1} \|g^i\|^2 \right)^\alpha} \quad \text{results in} \quad \mathbb{E} \|\nabla f(x^\tau)\| = \mathcal{O} \left( \frac{\max \left\{ \frac{n^{1/2}}{b^{3/4}}; 1 \right\}}{\sqrt{T}} \right). \tag{9}$$

*Here $\tau$ is chosen uniformly over $0, \dots, T-1$.*

Optimal choice of parameter $b$ is conducted as for L-SVRG above. After minimizing the expected number of gradient calls we end up with $b = n^{2/3}$.

**PAGE.** Next, we consider the PAGE method (Li et al., 2021a) - the loopless version of SARAH (Nguyen et al., 2017):

$$g^t = \begin{cases} \nabla f(x^t), & \text{with probability } p, \\ g^{t-1} + \frac{1}{b} \sum_{i \in S_t} \left( \nabla f_i(x^t) - \nabla f_i(x^{t-1}) \right), & \text{oth.} \end{cases} \tag{10}$$

where mini-batches $S_t$ of size $b$ are generated uniformly and independently at each iteration. With $p$ close to one, method is practically SGD, but with smaller probability it is similar to L-SVRG method, where mini-batches are used to correct the gradient estimation.

**Lemma 3.** *PAGE (10) satisfies Assumption 1 with* $\rho_1 = p, A = 0, B = \frac{1-p}{b}, \sigma_t^2 = 0, \rho_2 = 1, C = 0$.

**Corollary 3.** *In the non-convex case choosing step sizes as*

$$\gamma \lesssim \left( L \left[ 1 + \frac{1}{\sqrt{pb}} \right] \right)^{-1} \quad \text{results in} \quad \mathbb{E} \|\nabla f(x^\tau)\|^2 = \mathcal{O} \left( \frac{1}{T} \left( 1 + \frac{1}{\sqrt{pb}} \right) \right). \tag{11}$$

*Taking adaptive step sizes as*

$$\gamma_t = \frac{1}{\left(\max\left\{\frac{1}{\sqrt{pb}};1\right\}\right)^{1-\alpha}\left(\sum_{i=0}^{t-1}\|g^i\|^2\right)^\alpha} \quad \textit{results in} \quad \mathbb{E}\|\nabla f(x^\tau)\| = \mathcal{O}\left(\frac{\max\left\{\frac{1}{(pb)^{1/4}};1\right\}}{\sqrt{T}}\right). \quad (12)$$

*Here $\tau$ is chosen uniformly over $0,\ldots,T-1$.*

Minimizing the number of gradient calls for PAGE, we get $p = n^{-1/3}$ and $b = n^{2/3}$.

**ZeroSARAH.** Though, PAGE shows decent performance on various problems, the need to compute full gradients drastically increase the computation complexity. To deal with this, the ZeroSARAH algorithm (Li et al., 2021b) was proposed:

$$g^t = \frac{1}{b}\sum_{i\in S_t}[\nabla f_i(x^t) - \nabla f_i(x^{t-1})] + (1-\tfrac{b}{2n})g^{t-1} + \qquad y_i^{t+1} = \begin{cases} \nabla f_i(x^t), & i \in S_t, \\ y_i^t, & i \notin S_t, \end{cases}$$

$$+\frac{b}{2n}\left(\frac{1}{b}\sum_{i\in S_t}[\nabla f_i(x^{t-1}) - y_i^t] + \frac{1}{n}\sum_{j=1}^n y_j^t\right),$$

$$(13)$$

where mini-batches $S_t$ of size $b$ are generated uniformly and independently at each iteration.

**Lemma 4.** *ZeroSARAH (13) satisfies Assumption 1 with $\rho_1 = \frac{b}{2n}, A = \frac{b}{2n^2}, B = \frac{2}{b}, \sigma_t^2 = \frac{1}{n}\sum_{i=1}^n \|\nabla f_i(x^t) - y_i^t\|, \rho_2 = \frac{b}{2n}, C = \frac{2n}{b}$.*

**Corollary 4.** *In the non-convex case choosing step sizes as*

$$\gamma \lesssim \left(L\left[1 + \frac{\sqrt{n}}{b}\right]\right)^{-1} \quad \textit{results in} \quad \mathbb{E}\|\nabla f(x^\tau)\|^2 = \mathcal{O}\left(\frac{1}{T}\left(1 + \frac{\sqrt{n}}{b}\right)\right). \quad (14)$$

*Taking adaptive step sizes as*

$$\gamma_t = \frac{1}{\left(\max\left\{\frac{\sqrt{n}}{b};1\right\}\right)^{1-\alpha}\left(\sum_{i=0}^{t-1}\|g^i\|^2\right)^\alpha} \quad \textit{results in} \quad \mathbb{E}\|\nabla f(x^\tau)\| = \mathcal{O}\left(\frac{\max\left\{\frac{n^{1/4}}{b^{1/2}};1\right\}}{\sqrt{T}}\right). \quad (15)$$

*Here $\tau$ is chosen uniformly over $0,\ldots,T-1$.*

As for methods above one can compute the optimal batch size for ZeroSARAH, which equals to $b = n^{1/2}$.

By applying the novel unified assumption for existing algorithms we not only derive the same convergence rates for constant step sizes, as in original manuscripts (Defazio et al., 2014; Li et al., 2021a;b), but also show, that all method's adaptive variations obtain the same asymptotic $\mathcal{O}\left(1/\sqrt{T}\right)$, as non-adaptive. As shown in (Arjevani et al., 2023), this convergence rate is optimal in nonconvex setup, therefore, cannot be improved.

## 5.2 Distributed Optimization

In this section, we focus on distributed algorithms, that allow to reduce the amount of transmitted information between clients and server, while maintaining the overall convergence. We investigate following estimator schemes, such as EF-21 (Richtárik et al., 2021) and DASHA (Tyurin & Richtárik, 2022).

**EF-21.** Now that we have come to the distributed methods, we start with the definition of biased compressor

**Definition 3.** *Map $\mathcal{C} : \mathbb{R}^d \to \mathbb{R}^d$ is a biased compression operator, if there exist a constant $\delta \geq 1$, such that for all $x \in \mathbb{R}^d$*

$$\mathbb{E}[\|\mathcal{C}(x) - x\|^2] \leq \left(1 - \frac{1}{\delta}\right)\|x\|^2.$$

This is a broad class of compressors, that include greedy sparsifications, biased roundings and other operators. Though, simple compressing of the gradient do not lead to a demanded convergence, applying these operators to approximations' errors obtains better results. We start with the EF21

algorithm:

$$g_i^t = g_i^{t-1} + \mathcal{C}\left(\nabla f_i(x^t) - g_i^{t-1}\right), \qquad g^t = g^t + \frac{1}{n}\sum_{i=1}^{n}\mathcal{C}\left(\nabla f_i(x^t) - g_i^{t-1}\right). \qquad (16)$$

By compressing differences between true gradient and its estimation, this distributed method act as a variance reduction one. Biased compressor guarantees, that estimation error diminish throughout the iterations.

**Lemma 5.** *EF21 (16) satisfies Assumption 1 with* $\rho_1 = 1, A = 0, B = 0, \sigma_t^2 = \frac{1}{n}\sum_{i=1}^{n}\left\|g_i^t - \nabla f_i(x^t)\right\|^2, \rho_2 = \frac{1}{2\delta}, C = 2\delta.$

**Corollary 5.** *In the non-convex case choosing step sizes as*

$$\gamma \lesssim (L\left[1+\delta\right])^{-1} \quad \text{results in} \quad \mathbb{E}\left\|\nabla f(x^\tau)\right\|^2 = \mathcal{O}\left(\frac{1}{T}(1+\delta)\right). \qquad (17)$$

*Taking adaptive step sizes as*

$$\gamma_t = \frac{1}{\delta^{1-\alpha}\left(\sum_{i=0}^{t-1}\|g^i\|^2\right)^\alpha} \quad \text{results in} \quad \mathbb{E}\left\|\nabla f(x^\tau)\right\| = \mathcal{O}\left(\frac{\delta^{1/2}}{\sqrt{T}}\right). \qquad (18)$$

*Here* $\tau$ *is chosen uniformly over* $0,\ldots,T-1.$

**DASHA.** Besides biased compressors, unbiased once are also utilized in distributed optimization

**Definition 4.** *Map* $\mathcal{Q} : \mathbb{R}^d \to \mathbb{R}^d$ *is an unbiased compression operator, if there exist a constant* $\omega \geq 1$ *such that for all* $x \in \mathbb{R}^d$
$$\mathbb{E}\mathcal{Q}(x) = x, \qquad \mathbb{E}[\|\mathcal{Q}(x)\|^2] \leq \omega\|x\|^2.$$

This class of compressors include such operators, as unbiased sparsifications, roundings and others. One of advantages over biased compressors is that these do not change the vector in mean, that might lead to a better convergence rates with the growing number of nodes.

To utilize unbiased compressors, one may consider changing the finite sum methods, by replacing the batch averaging with the quantization. However, most derived variations might suffer from transmitting full gradient which is present in PAGE, for instance. To overcome this obstacle, DASHA algorithm was proposed (Tyurin & Richtárik, 2022), that incorporates momentum to get rid of transferring the uncompressed vectors.

$$\Delta_i^t = \mathcal{Q}\left(\nabla f_i(x^t) - \nabla f_i(x^{t-1}) - \frac{1}{2\omega+1}\left(g_i^{t-1} - \nabla f_i(x^t)\right)\right)$$
$$g_i^t = g_i^{t-1} + \Delta_i^t, \qquad g^t = g^t + \frac{1}{n}\sum_{i=1}^{n}\Delta_i^t. \qquad (19)$$

**Lemma 6.** *DASHA (19) satisfies Assumption 1 with* $\rho_1 = \frac{1}{2\omega+1}, A = \frac{2\omega}{(2\omega+1)^2 n}, B = \frac{2\omega}{n}, \sigma_t^2 = \frac{1}{n}\sum_{i=1}^{n}\left\|g_i^t - \nabla f_i(x^t)\right\|^2, \rho_2 = \frac{1}{2\omega+1}, C = 2\omega.$

**Corollary 6.** *In the non-convex case choosing step sizes as*

$$\gamma \lesssim \left(L\left[1+\frac{\omega}{\sqrt{n}}\right]\right)^{-1} \quad \text{results in} \quad \mathbb{E}\left\|\nabla f(x^\tau)\right\|^2 = \mathcal{O}\left(\frac{1}{T}\left(1+\frac{\omega}{\sqrt{n}}\right)\right). \qquad (20)$$

*Taking adaptive step sizes as*

$$\gamma_t = \frac{1}{\left(\max\left\{\frac{\omega}{\sqrt{n}};1\right\}\right)^{1-\alpha}\left(\sum_{i=0}^{t-1}\|g^i\|^2\right)^\alpha} \quad \text{results in} \quad \mathbb{E}\left\|\nabla f(x^\tau)\right\| = \mathcal{O}\left(\frac{\max\left\{\frac{\omega^{1/2}}{n^{1/4}};1\right\}}{\sqrt{T}}\right). \qquad (21)$$

*Here* $\tau$ *is chosen uniformly over* $0,\ldots,T-1.$

We have shown, that various distributed optimization algorithms can be described not only with proposed unified scheme, but also be implemented with adaptive step sizes. To our knowledge, these are first distributed adaptive algorithms, which are, moreover, parameter-free. Adaptive algorithms' variations have the same asymptotic $\mathcal{O}\left(1/\sqrt{T}\right)$, as non-adaptive. It is optimal in non-convex scenario and cannot be improved.

## 5.3 OTHER METHODS

More algorithms for finite sum problems, distributed optimization, as well as coordinate methods with new established adaptive variations can be found in Appendix due to space limitation.

## 6 NUMERICAL EXPERIMENTS

We validate the performance of the proposed adaptive methods on the logistic regression problem: $\min_{x \in \mathbb{R}^d} \left\{ f(x) = \sum_{i=1}^{n} \log \left( 1 + \exp \left( -b_i \cdot x^T a_i \right) \right) \right\}$, where $x$ are model weights and $\{a_i, b_i\}$ are training samples with $a_i \in \mathbb{R}^d, b_i \in \{-1, 1\}$. Experiments use the LibSVM dataset a9a (Chang & Lin, 2011).

We compare our methods against their theoretical and best-tuned stepsize versions. Theoretical stepsizes follow the original papers, with smoothness constant $L$ estimated as the largest Hessian eigenvalue. For our methods, we set $\alpha = 0.33$, the least robust value in training. In the finite-sum setting, we compare SAGA (7), PAGE (10), and ZeroSARAH (13) with their parameter-free counterparts: PFSAGA (7+9), PFPAGE (10+12), and PFZeroSARAH (13+15). For SAGA/PFSAGA we use $b \sim n^{2/3}$; for PAGE/PFPAGE, the same batch size with $p = n^{-1/3}$; and for ZeroSARAH/PFZeroSARAH, $b = n^{1/2}$. Performance is reported in iterations vs. gradient norm, with Adam (batch size $b \sim n^{2/3}$, tuned learning rate) as baseline. In the distributed setting, we compare EF21 (16) with its parameter-free variant PFEF21 (16+18). We use 10 clients and TopK compression (Alistarh et al., 2018), selecting the top $k = 0.05d$ coordinates by magnitude. Further compression results appear in the Appendix. The plots show that our proposed methods outperform

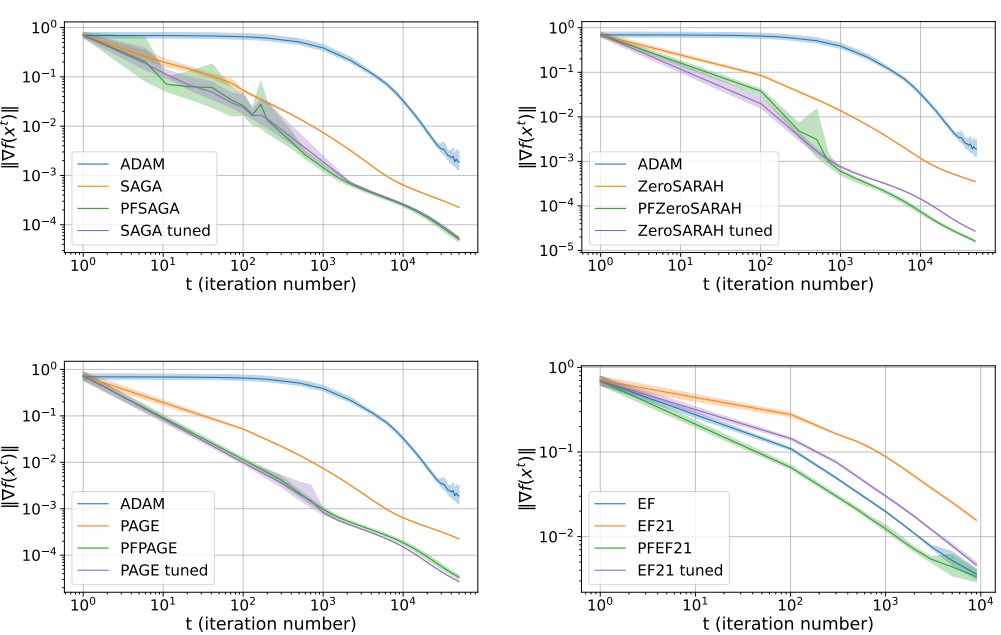

Figure 1: Results on the a9a dataset showing convergence behaviour of SAGA, PAGE, ZeroSARAH and EF21 with theoretical, tuned and adaptive stepsize.

those using both theoretical and tuned step sizes. Notably, the parameter-free variants require no tuning, making them a more practical and appealing choice.

It can be noticed, that Adam do not outperform variance-reduced methods. Actually, this is not surprising for several problems. Discussion of this phenomenon can be found in the blog[1]. Additional results, that compare more methods can be found in Appendix.

---

[1]https://parameterfree.com/2020/12/06/neural-network-maybe-evolved-to-make-adam-the-best-optimizer

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

## A    Comparison to Previous Unified Analyses

As unified analyses explain the convergence properties of wide range of methods, they became especially popular in the last years. Derived for composite problems (Khaled et al., 2023), variational inequalities (Beznosikov et al., 2023a), asynchronous optimization (Islamov et al., 2024), these analyses are extended to various problem statements. However, the key element is the analysis of SGD and related methods. Below we specify the distinctions between existing approaches and our proposed adaptive mechanism.

Vanilla unified SGD analysis was suggested in (Stich, 2019), where strongly convex functions were considered. The assumptions on stochastic gradient $g^t$ were the following:

$$\mathbb{E}\left[g^t \mid \mathcal{F}_t\right] = \nabla f(x^t). \tag{22}$$

$$\mathbb{E}\left[\|g^t\|^2 \mid \mathcal{F}_t\right] \le 2L(f(x^t) - f_*) + \sigma^2, \tag{23}$$

where $L$ stands for smoothness constant. The stepsizes were taken as $\gamma_t \le \frac{1}{2L}$. This results in $\mathcal{O}\left(LR^2 \exp\left[-\frac{\mu T}{4L}\right] + \frac{\sigma^2}{\mu T}\right)$, where $R = \|x^0 - x^*\|$. It can be seen, that without diminishable terms, convergence is not optimal in strongly convex setup.

This suboptimality was addressed in (Gorbunov et al., 2020), where the strongly quasi-convex is analyzed. They also demand unbiased stochastic gradient, but with the utilization of variance reduction scheme, they obtain diminishing terms on error approximation bound. The considered assumptions are:

$$\mathbb{E}\left[g^t \mid x^t\right] = \nabla f(x^t). \tag{24}$$

$$\mathbb{E}\left[\|g^t\|^2 \mid x^t\right] \quad \le \quad 2A\left(f(x^t) - f_*\right) + B\sigma_t^2, \tag{25}$$

$$\mathbb{E}\left[\sigma_{t+1}^2 \mid \sigma_t^2\right] \quad \le \quad (1 - \rho)\sigma_t^2 + C(f(x^t) - f_*), \tag{26}$$

where $\{\sigma_t\}_{t \ge 0}$ is a (possibly) random sequence. They introduce and analyze the Lyapunov function $V^t = \|x^t - x_*\| + \frac{2B}{\rho}\gamma^2\sigma_t^2$ for constant stepsize $\gamma = \min\left\{\frac{1}{\mu}, \frac{1}{A + 2BC/\rho}\right\}$. This allows to obtain a linear convergence: $\mathcal{O}\left(\max\left\{(1 - \gamma\mu)^t, \left(1 - \frac{\rho}{2}\right)^t\right\} V^0\right)$. However, this analysis still requires constant stepsize and strong assumptions on strongly quasi-convexity.

The approach with Lyapunov function turned out to be productive, as it also was incorporated in nonconvex analyses (Khaled et al., 2023; Li & Richtárik, 2020). But they still required the unbiased gradient estimator:

$$\mathbb{E}\left[g^t \mid x^t\right] = \nabla f(x^t). \tag{27}$$

$$\mathbb{E}\left[\|g^t\|^2 \mid x^t\right] \quad \le \quad 2A_1\left(f(x^t) - f_*\right) + B\|\nabla f(x^t)\|^2 + C_1\sigma_t^2, \tag{28}$$

$$\mathbb{E}\left[\sigma_{t+1}^2 \mid x^t\right] \quad \le \quad (1 - \rho)\sigma_t^2 + 2A_2\left(f(x^t) - f_*\right) + B_2\|\nabla f(x^t)\|^2. \tag{29}$$

These assumptions are more flexible, but the analysis was conducted for constant stepsize $\gamma = \frac{1}{LB_1 + LC_1B_2\rho^{-1}}$. In (Li & Richtárik, 2020) they also analyzed the stepsize, if the number of iterations was known beforehand, but the stepsize was still constant.

One of the attempts to get rid of the gradient unbiasedness was done in (Driggs et al., 2022), where they suggested *memory-biased* gradient estimators:

$$\mathbb{E}g^t - \nabla f(x^t) = \left(1 - \frac{1}{\theta}\right)\left(\frac{1}{n}\sum_{i=1}^n \nabla f_i(\varphi_i^t) - \nabla f(x^t)\right), \tag{30}$$

and *recursively-biased* gradient estimators:

$$\mathbb{E}g^t - \nabla f(x^t) = (1 - \rho_B)(g^{t-1} - \nabla f(x^{t-1})). \tag{31}$$

These assumptions were sufficient to derive optimal convergence in convex setup. However, constant step size was considered, that depends on problem's properties.

Our Assumption 1 does not require neither unbiased, nor recursively- or memory- biased gradients for nonconvex setup:

$$\mathbb{E}\left[\|g^t - \nabla f(x^t)\|^2 \mid \mathcal{F}_t\right] \quad \le \quad (1 - \rho_1)\|g^{t-1} - \nabla f(x^{t-1})\|^2 + A\sigma_{t-1}^2 + BL^2\|x^t - x^{t-1}\|^2,$$

$$\mathbb{E}\left[\sigma_t^2 \mid \mathcal{F}_t\right] \quad \le \quad (1 - \rho_2)\sigma_{t-1}^2 + CL^2\|x^t - x^{t-1}\|^2. \tag{32}$$

The class of appropriate estimators increases, the convergence rates stays the same (optimal), and the adaptive stepsizes can be utilized.

## B    CONVERGENCE GUARANTEES

### B.1    NON-CONVEX CASE

**Lemma 7.** *(Lemma 2 from Li et al. (2021a)) Let $f$ be $L$-smooth, then iteration of Algorithm 1 satisfies*

$$f(x^{t+1}) \leq f(x^t) - \frac{\gamma_t}{2} \left\| \nabla f(x^t) \right\|^2 + \left( \frac{L}{2} - \frac{1}{2\gamma} \right) \left\| x^{t+1} - x^t \right\|^2 + \frac{\gamma}{2} \left\| g^t - \nabla f(x^t) \right\|^2$$

**Theorem 4** (Non-convex convergence). *Let $f$ be $L$-smooth and satisfy Assumption 1. Then Algorithm 1 with step size*

$$\gamma_t \equiv \gamma \leq \frac{1}{L} \left( 1 + \sqrt{\frac{B\rho_2 + AC}{\rho_1 \rho_2}} \right)^{-1},$$

*for any $T > 0$ achieves*

$$\frac{1}{T} \sum_{t=0}^{T-1} \mathbb{E} \| \nabla f(x^t) \|^2 \leq \frac{2V^0}{\gamma T},$$

*where $V^0 = f(x^0) - f_* + \frac{\gamma}{2\rho_1} \left\| g^0 - \nabla f(x^0) \right\|^2 + \frac{\gamma A}{2\rho_1 \rho_2} \sigma_0^2$.*

*Proof.*

$$f(x^{t+1}) - f_* \quad \leq \quad f(x^t) - f_* - \frac{\gamma}{2} \left\| \nabla f(x^t) \right\|^2 + \left( \frac{L}{2} - \frac{1}{2\gamma} \right) \left\| x^{t+1} - x^t \right\|^2 + \frac{\gamma}{2} \left\| g^t - \nabla f(x^t) \right\|^2.$$

Add $\mu G^{t+1} := \mu \left\| g^{t+1} - \nabla f(x^{t+1}) \right\|^2$, $\theta \sigma_{t+1}^2$, and take conditional expectation. Define $\delta_t = f(x^t) - f_*$ and $r_t = \| x^{t+1} - x^t \|^2$. Hence,

$$\mathbb{E}_{\xi_t} \left[ \delta^{t+1} + \mu G^{t+1} + \theta \sigma_{t+1}^2 \right] \quad \leq \quad \delta^t - \frac{\gamma}{2} \left\| \nabla f(x^t) \right\|^2 + \left( \frac{L}{2} - \frac{1}{2\gamma} + \mu B L^2 + \theta C L^2 \right) r_t^2$$

$$+ \quad \left( \frac{\gamma}{2} + \mu(1 - \rho_1) \right) G^t + (\mu A + \theta(1 - \rho_2)) \sigma_t^2.$$

Set $\mu = \frac{\gamma}{2\rho_1}$, $\theta = \frac{\gamma A}{2\rho_1 \rho_2}$, therefore with $\gamma^2 L^2 \frac{B\rho_2 + AC}{\rho_1 \rho_2} + \gamma L \leq 1$:

$$\mathbb{E} \left[ \delta^{t+1} + \frac{\gamma}{2\rho_1} G^{t+1} + \frac{\gamma A}{2\rho_1 \rho_2} \sigma_{t+1}^2 \right] \leq \mathbb{E} \left[ \delta^t + \frac{\gamma}{2\rho_1} G^t + \frac{\gamma A}{2\rho_1 \rho_2} \sigma_t^2 - \frac{\gamma}{2} \left\| \nabla f(x^t) \right\|^2 \right],$$

and

$$\frac{1}{T} \sum_{t=0}^{T-1} \mathbb{E} \| \nabla f(x^t) \|^2 \leq \frac{2V^0}{\gamma T}.$$

$\square$

### B.2    PL CASE

**Theorem 5** (PL convergence). *Let $f$ be $L$-smooth, satisfy PL condition and Assumption 1. Then, Algorithm 1 with step size*

$$\gamma_t \equiv \gamma \leq \min \left\{ \frac{1}{L} \left( 1 + \sqrt{\frac{B\rho_2 + 4AC}{\rho_1 \rho_2}} \right)^{-1}, \frac{\min\{\rho_1, \rho_2\}}{2\mu} \right\},$$

*for any $T > 0$ achieves*

$$V^T \leq (1 - \gamma\mu)^T V^0,$$

*where $V^t = f(x^t) - f_* + \frac{\gamma}{\rho_1} \left\| g^t - \nabla f(x^t) \right\|^2 + \frac{2\gamma A}{\rho_1 \rho_2} \sigma_t^2$.*

*Proof.*

$$f(x^{t+1}) - f_* \quad \leq \quad f(x^t) - f_* - \frac{\gamma}{2} \left\| \nabla f(x^t) \right\|^2 + \left( \frac{L}{2} - \frac{1}{2\gamma} \right) \left\| x^{t+1} - x^t \right\|^2 + \frac{\gamma}{2} \left\| g^t - \nabla f(x^t) \right\|^2.$$

Add $\mu G^{t+1} := \mu \left\| g^{t+1} - \nabla f(x^{t+1}) \right\|^2$, $\theta\sigma_{t+1}^2$, and take conditional expectation. Define $\delta_t = f(x^t) - f_*$ and $r_t = \|x^{t+1} - x^t\|^2$. Hence,

$$
\begin{aligned}
\mathbb{E}_{\xi_t}\left[\delta^{t+1} + \mu G^{t+1} + \theta\sigma_{t+1}^2\right] &\leq \delta^t - \frac{\gamma}{2}\left\|\nabla f(x^t)\right\|^2 + \left(\frac{L}{2} - \frac{1}{2\gamma} + \mu BL^2 + \theta CL^2\right)r_t^2 \\
&\quad + \left(\frac{\gamma}{2} + \mu(1-\rho_1)\right)G^t + (\mu A + \theta(1-\rho_2))\sigma_t^2.
\end{aligned}
$$

From PL condition we have:

$$
\mathbb{E}_{\xi_t}\left[\delta^{t+1} + \mu G^{t+1} + \theta\sigma_{t+1}^2\right] \leq (1-\gamma\mu)\delta^t + \left(\frac{L}{2} - \frac{1}{2\gamma} + \mu B + \theta C\right)r_t^2 + \left(\frac{\gamma}{2} + \mu(1-\rho_1)\right)G^t + (\mu A + \theta(1-\rho_2))\sigma_t^2.
$$

Set $\mu = \frac{\gamma}{\rho_1}$, $\theta = \frac{2\gamma A}{\rho_1 \rho_2}$, therefore with $\gamma^2\frac{B\rho_2+4AC}{\rho_1\rho_2} + \gamma L \leq 1$:

$$
\mathbb{E}_{\xi_t}\left[\delta^{t+1} + \frac{\gamma}{\rho_1}G^{t+1} + \frac{2\gamma A}{\rho_1\rho_2}\sigma_{t+1}^2\right] \leq (1-\gamma\mu)\delta^t + \frac{\gamma}{2\rho_1}\left(1 - \frac{\rho_1}{2}\right)G^t + \frac{2\gamma A}{\rho_1\rho_2}\left(1 - \frac{\rho_2}{2}\right)\sigma_t^2,
$$

Therefore, if $\gamma \leq \frac{\min\{\rho_1,\rho_2\}}{2\mu}$, then

$$
V^{t+1} \leq (1-\gamma\mu)V^t \leq (1-\gamma\mu)^{t+1}V^0.
$$

$\square$

### B.3 ADAPTIVE STEP SIZES

**Lemma 8.** *Suppose $c_i$ is positive for every $i$ and let $0 < \alpha < 1$. We can ensure that*

$$
\left(\sum_{i=1}^n c_i\right) \leq \sum_{i=1}^n \frac{c_i}{\left(\sum_{j=1}^i c_j\right)^\alpha} \leq \frac{1}{1-\alpha}\left(\sum_{i=1}^n c_i\right)^{1-\alpha}
$$

**Lemma 9** (Decent lemma I). *Let $\gamma_{t+1} \leq \gamma_t$ and $\gamma_t \in \mathcal{F}_t = \sigma(x_0, \ldots, x_t)$. Define $V_t = \mathbb{E}\left[f(x_t) + \frac{\gamma_t}{2\rho_1}\|g_t - \nabla f(x_t)\|^2 + \frac{\gamma_t A}{2\rho_1\rho_2}\sigma_t^2\right]$. Then, we can derive*

$$
\mathbb{E}\sum_{t=0}^{T-1}\gamma_t\|g_t\|^2 \leq 2V_0 + L\mathbb{E}\sum_{t=0}^{T-1}\gamma_t^2\|g_t\|^2 + \frac{B\rho_2 + AC}{\rho_1\rho_2}L^2\mathbb{E}\sum_{t=0}^{T-1}\gamma_t^3\|g_t\|^2
$$

*Proof.* From $L$-smoothness we have

$$
f(x_{t+1}) \leq f(x_t) - \frac{\gamma_t}{2}\|\nabla f(x_t)\|^2 - \frac{\gamma_t}{2}\|g_t\|^2 + \frac{\gamma_t}{2}\|g_t - \nabla f(x_t)\|^2 + \frac{L\gamma_t^2}{2}\|g_t\|^2.
$$

Take expectation and add $\mathbb{E}\left[\frac{\gamma_t}{2\rho_1}\|g_{t+1} - \nabla f(x_{t+1})\|^2 + \frac{\gamma_t A}{2\rho_1\rho_2}\sigma_{t+1}^2 \mid \mathcal{F}_t\right]$. Then,

$$
\begin{aligned}
\mathbb{E}\left[f(x_{t+1}) + \frac{\gamma_t}{2\rho_1}\|g_{t+1} - \nabla f(x_{t+1})\|^2 + \frac{\gamma_t A}{2\rho_1\rho_2}\sigma_{t+1}^2 \mid \mathcal{F}_t\right] &\leq f(x_t) - \frac{\gamma_t}{2}\|\nabla f(x_t)\|^2 - \frac{\gamma_t}{2}\|g_t\|^2 \\
+ \frac{\gamma_t}{2\rho_1}\|g_t - \nabla f(x_t)\|^2 + \frac{\gamma_t A}{2\rho_1\rho_2}\sigma_t^2 + \frac{B\rho_2 + AC}{2\rho_1\rho_2}\gamma_t^3\|g_t\|^2 &+ \frac{L\gamma_t^2}{2}\|g_t\|^2.
\end{aligned}
$$

Use the fact, that $\gamma_{t+1} \leq \gamma_t$, then

$$
\begin{aligned}
\mathbb{E}\left[f(x_{t+1}) + \frac{\gamma_{t+1}}{2\rho_1}\|g_{t+1} - \nabla f(x_{t+1})\|^2 + \frac{\gamma_{t+1} A}{2\rho_1\rho_2}\sigma_{t+1}^2 \mid \mathcal{F}_t\right] &\leq f(x_t) - \frac{\gamma_t}{2}\|\nabla f(x_t)\|^2 - \frac{\gamma_t}{2}\|g_t\|^2 \\
+ \frac{\gamma_t}{2\rho_1}\|g_t - \nabla f(x_t)\|^2 + \frac{\gamma_t A}{2\rho_1\rho_2}\sigma_t^2 + \frac{B\rho_2 + AC}{2\rho_1\rho_2}\gamma_t^3\|g_t\|^2 &+ \frac{L\gamma_t^2}{2}\|g_t\|^2.
\end{aligned}
$$

Take full expectation on both sides, sum up and multiply by 2. Hence,

$$
\sum_{t=1}^T 2V_t \leq \sum_{t=0}^{T-1} 2V_t - \mathbb{E}\gamma_t\|\nabla f(x_t)\|^2 - \mathbb{E}\gamma_t\|g_t\|^2 + \frac{B\rho_2 + AC}{\rho_1\rho_2}\mathbb{E}\gamma_t^3\|g_t\|^2 + L\mathbb{E}\gamma_t^2\|g_t\|^2.
$$

After rearranging the terms:

$$
\mathbb{E}\sum_{t=0}^{T-1}\gamma_t\|g_t\|^2 \leq 2V_0 + L\mathbb{E}\sum_{t=0}^{T-1}\gamma_t^2\|g_t\|^2 + \frac{B\rho_2 + AC}{\rho_1\rho_2}\mathbb{E}\sum_{t=0}^{T-1}\gamma_t^3\|g_t\|^2
$$

$\square$

**Lemma 10.** *With the choice $\gamma_t = \frac{1}{\nu^{\frac{1-\alpha}{2}}\left(\sum\limits_{i=1}^{t}\|g_i\|^2\right)^{\alpha}}$, we have*

$$\sum_{t=0}^{T-1}\mathbb{E}\|g_t\| \leq G^{\frac{1}{2(1-\alpha)}}\nu^{\frac{1}{4}},$$

*where* $G = 6V_0 + \frac{3\alpha}{1-\alpha}\left(\frac{1}{1-2\alpha}\left(\frac{3-6\alpha}{1-\alpha}\right)^{\frac{1-2\alpha}{1-\alpha}}\right)^{\frac{1-\alpha}{\alpha}}\left(\frac{L}{\sqrt{\nu}}\right)^{\frac{1-\alpha}{\alpha}} + \frac{6\alpha}{1-\alpha}\left(\frac{1}{1-3\alpha}\left(\frac{3-9\alpha}{1-\alpha}\right)^{\frac{1-3\alpha}{1-\alpha}}\right)^{\frac{1-3\alpha}{2\alpha}}$

$\cdot \left(\frac{B\rho_2+AC}{\rho_1\rho_2\nu}\right)^{\frac{1-\alpha}{2\alpha}}$

*Proof.* According to Lemma 8, we have

$$\sum_{t=0}^{T-1}\gamma_t\|g_t\|^2 = \sum_{t=0}^{T-1}\frac{\|g_t\|^2}{\nu^{\frac{1-\alpha}{2}}\left(\sum\limits_{i=1}^{t}\|g_i\|^2\right)^{\alpha}} \geq \left(\frac{1}{\sqrt{\nu}}\sum_{t=0}^{T-1}\|g_t\|^2\right)^{1-\alpha}.$$

Then, we aim to bound $L\sum\limits_{t=0}^{T-1}\gamma_t^2\|g_t\|^2$ and $\frac{B\rho_2+AC}{\rho_1\rho_2}\sum\limits_{t=0}^{T-1}\gamma_t^3\|g_t\|^2$. For the first term we have

$$
\begin{aligned}
L\sum_{t=0}^{T-1}\gamma_t^2\|g_t\|^2 &= L\sum_{t=0}^{T-1}\frac{\|g_t\|^2}{\nu^{\frac{2-2\alpha}{2}}\left(\sum\limits_{i=0}^{t}\|g_i\|^2\right)2\alpha} = \frac{L}{\sqrt{\nu}}\sum_{t=0}^{T-1}\frac{\|g_t\|^2}{\nu^{\frac{1-2\alpha}{2}}\left(\sum\limits_{i=0}^{t}\|g_i\|^2\right)2\alpha} \\
&\leq \frac{L}{\sqrt{\nu}}\frac{1}{1-2\alpha}\left(\frac{1}{\sqrt{\nu}}\sum_{t=0}^{T-1}\|g_t\|^2\right)^{1-2\alpha} \\
&= \frac{L}{\sqrt{\nu}}\frac{1}{1-2\alpha}\left(\frac{3-6\alpha}{1-\alpha}\right)^{\frac{1-2\alpha}{1-\alpha}}\left(\frac{1-\alpha}{3-6\alpha}\right)^{\frac{1-2\alpha}{1-\alpha}}\left(\frac{1}{\sqrt{\nu}}\sum_{t=0}^{T-1}\|g_t\|^2\right)^{1-2\alpha} \\
&\leq G_1\left(\frac{L}{\sqrt{\nu}}\right)^{\frac{1-\alpha}{\alpha}} + \frac{1}{3}\left(\frac{1}{\sqrt{\nu}}\sum_{t=0}^{T-1}\|g_t\|^2\right)^{1-\alpha},
\end{aligned}
$$

where $G_1 = \frac{\alpha}{1-\alpha}\left(\frac{1}{1-2\alpha}\left(\frac{3-6\alpha}{1-\alpha}\right)^{\frac{1-2\alpha}{1-\alpha}}\right)^{\frac{1-\alpha}{\alpha}}$. For another term we similarly achieve

$$
\begin{aligned}
\frac{B\rho_2+AC}{\rho_1\rho_2}\sum_{t=0}^{T-1}\gamma_t^3\|g_t\|^2 &= \frac{B\rho_2+AC}{\rho_1\rho_2}\sum_{t=0}^{T-1}\frac{\|g_t\|^2}{\nu^{\frac{3-3\alpha}{2}}\left(\sum\limits_{i=0}^{t}\|g_i\|^2\right)3\alpha} = \frac{B\rho_2+AC}{\rho_1\rho_2\nu}\sum_{t=0}^{T-1}\frac{\|g_t\|^2}{\nu^{\frac{1-3\alpha}{2}}\left(\sum\limits_{i=0}^{t}\|g_i\|^2\right)3\alpha} \\
&\leq \frac{B\rho_2+AC}{\rho_1\rho_2\nu}\frac{1}{1-3\alpha}\left(\frac{1}{\sqrt{\nu}}\sum_{t=0}^{T-1}\|g_t\|^2\right)^{1-3\alpha} \\
&= \frac{B\rho_2+AC}{\rho_1\rho_2\nu}\left(\frac{3-9\alpha}{1-\alpha}\right)^{\frac{1-3\alpha}{1-\alpha}}\left(\frac{1-\alpha}{3-9\alpha}\right)^{\frac{1-3\alpha}{1-\alpha}}\frac{1}{1-3\alpha}\left(\frac{1}{\sqrt{\nu}}\sum_{t=0}^{T-1}\|g_t\|^2\right)^{1-3\alpha} \\
&\leq G_2\left(\frac{B\rho_2+AC}{\rho_1\rho_2\nu}\right)^{\frac{1-\alpha}{2\alpha}} + \frac{1}{3}\left(\frac{1}{\sqrt{\nu}}\sum_{t=0}^{T-1}\|g_t\|^2\right)^{1-\alpha},
\end{aligned}
$$

where $G_2 = \frac{2\alpha}{1-\alpha} \left( \frac{1}{1-3\alpha} \left( \frac{3-9\alpha}{1-\alpha} \right)^{\frac{1-3\alpha}{1-\alpha}} \right)^{\frac{1-\alpha}{2\alpha}}$. After taking expectation and applying the results of Lemma 9, we achieve

$$\mathbb{E} \left( \frac{1}{\sqrt{\nu}} \sum_{t=0}^{T-1} \|g_t\|^2 \right)^{1-\alpha} \leq 2V_0 + G_1 \left( \frac{L}{\sqrt{\nu}} \right)^{\frac{1-\alpha}{\alpha}} + G_2 \left( \frac{B\rho_2 + AC}{\rho_1 \rho_2 \nu} \right)^{\frac{1-\alpha}{2\alpha}} + \frac{2}{3} \mathbb{E} \left( \frac{1}{\sqrt{\nu}} \sum_{t=0}^{T-1} \|g_t\|^2 \right)^{1-\alpha},$$

$$\mathbb{E} \left( \frac{1}{\sqrt{\nu}} \sum_{t=0}^{T-1} \|g_t\|^2 \right)^{1-\alpha} \leq 6V_0 + 3G_1 \left( \frac{L}{\sqrt{\nu}} \right)^{\frac{1-\alpha}{\alpha}} + 3G_2 \left( \frac{B\rho_2 + AC}{\rho_1 \rho_2 \nu} \right)^{\frac{1-\alpha}{2\alpha}}.$$

Utilize the Holder's inequalities and using $\alpha < 1/3$ we acquire

$$\mathbb{E} \frac{1}{T} \sum_{t=0}^{T-1} \|g_t\| \leq \mathbb{E} \left( \frac{1}{T} \sum_{t=0}^{T-1} \|g_t\|^2 \right)^{1/2}$$

$$\mathbb{E} \left( \frac{1}{\sqrt{\nu}} \sum_{t=0}^{T-1} \|g_t\|^2 \right)^{1/2} \leq \left( \mathbb{E} \left( \frac{1}{\sqrt{\nu}} \sum_{t=0}^{T-1} \|g_t\|^2 \right)^{1-\alpha} \right)^{\frac{1}{2(1-\alpha)}} \leq \left( 6V_0 + 3G_1 \left( \frac{L}{\sqrt{\nu}} \right)^{\frac{1-\alpha}{\alpha}} + 3G_2 \left( \frac{B\rho_2 + AC}{\rho_1 \rho_2 \nu} \right)^{\frac{1-\alpha}{2\alpha}} \right)^{\frac{1}{2(1-\alpha)}}.$$

Therefore,

$$\frac{1}{T} \sum_{t=0}^{T-1} \mathbb{E} \|g_t\| \leq G^{\frac{1}{2(1-\alpha)}} \nu^{\frac{1}{4}}.$$

$\square$

**Lemma 11** (Decent lemma II). *We have*

$$\mathbb{E} \sum_{t=0}^{T-1} \|g_t - \nabla f(x_t)\|^2 \leq \left( 1 + \frac{1}{\rho_1} \right) \|g_0 - \nabla f(x_0)\|^2 + \frac{A}{\rho_1} \left( 1 + \frac{1}{\rho_2} \right) \sigma_0^2 + \frac{B\rho_2 + AC}{\rho_1 \rho_2} L^2 \mathbb{E} \sum_{t=0}^{T-1} \gamma_t^2 \|g_t\|^2$$

*Proof.* From assumptions we have

$$\mathbb{E} \sum_{t=0}^{T-1} \|g_{t+1} - \nabla f(x_{t+1})\|^2 \leq (1-\rho_1) \mathbb{E} \sum_{t=0}^{T-1} \|g_t - \nabla f(x_t)\|^2 + A\mathbb{E} \sum_{t=0}^{T-1} \sigma_t^2 + BL^2 \mathbb{E} \sum_{t=0}^{T-1} \gamma_t^2 \|g_t\|^2$$

$$\mathbb{E} \|g_T - \nabla f(x_T)\|^2 + \rho_1 \mathbb{E} \sum_{t=1}^{T-1} \|g_t - \nabla f(x_t)\|^2 \leq \|g_0 - \nabla f(x_0)\|^2 + A\mathbb{E} \sum_{t=0}^{T-1} \sigma_t^2 + BL^2 \mathbb{E} \sum_{t=0}^{T-1} \gamma_t^2 \|g_t\|^2$$

Similarly for $\sigma_t^2$ we obtain

$$\mathbb{E} \sum_{t=0}^{T-1} \sigma_{t+1}^2 \leq (1-\rho_2) \mathbb{E} \sum_{t=0}^{T-1} \sigma_t^2 + CL^2 \mathbb{E} \sum_{t=0}^{T-1} \gamma_t^2 \|g_t\|^2$$

$$\mathbb{E} \sigma_T^2 + \rho_2 \mathbb{E} \sum_{t=1}^{T-1} \sigma_t^2 \leq \sigma_0^2 + CL^2 \mathbb{E} \sum_{t=0}^{T-1} \gamma_t^2 \|g_t\|^2.$$

Combining all these inequalities we obtain

$$\mathbb{E} \sum_{t=1}^{T-1} \|g_t - \nabla f(x_t)\|^2 \leq \frac{1}{\rho_1} \left( \|g_0 - \nabla f(x_0)\|^2 + A\mathbb{E} \sum_{t=0}^{T-1} \sigma_t^2 + BL^2 \mathbb{E} \sum_{t=0}^{T-1} \gamma_t^2 \|g_t\|^2 \right)$$

$$\leq \frac{1}{\rho_1} \left( \|g_0 - \nabla f(x_0)\|^2 + A\sigma_0^2 + \frac{A}{\rho_2} \sigma_0^2 + \frac{AC}{\rho_2} L^2 \mathbb{E} \sum_{t=0}^{T-1} \gamma_t \|g_t\|^2 + BL^2 \mathbb{E} \sum_{t=0}^{T-1} \gamma_t^2 \|g_t\|^2 \right).$$

Add $\mathbb{E} \|g_0 - \nabla f(x_0)\|^2$, hence

$$\mathbb{E} \sum_{t=0}^{T-1} \|g_t - \nabla f(x_t)\|^2 \leq \left( 1 + \frac{1}{\rho_1} \right) \|g_0 - \nabla f(x_0)\|^2 + \frac{A}{\rho_1} \left( 1 + \frac{1}{\rho_2} \right) \sigma_0^2 + \frac{B\rho_2 + AC}{\rho_1 \rho_2} L^2 \mathbb{E} \sum_{t=0}^{T-1} \gamma_t^2 \|g_t\|^2.$$

$\square$

**Lemma 12.** *With the choice* $\gamma_t = \dfrac{1}{\nu^{\frac{1-\alpha}{2}} \left( \sum\limits_{i=1}^{t} \|g_i\|^2 \right)^{\alpha}}$, *we have*

$$\sum_{t=0}^{T-1} \mathbb{E}\|g_t - \nabla f(x_t)\|^2 \leq \left(1 + \frac{1}{\rho_1}\right) \|g_0 - \nabla f(x_0)\|^2 + \frac{A}{\rho_1}\left(1 + \frac{1}{\rho_2}\right)\sigma_0^2 + H_1 + H_2 \mathbb{E}\left(\sum_{t=0}^{T-1} \|g_t\|^2\right)^{1-\alpha},$$

*where* $H_1 = \frac{\alpha}{1-\alpha}\left(\frac{1}{1-2\alpha}\left(\frac{2-4\alpha}{1-\alpha}\right)^{\frac{1-2\alpha}{1-\alpha}}\right)^{\frac{1-\alpha}{\alpha}}\left(\frac{B\rho_2+AC}{\rho_1\rho_2}\nu^{1-\alpha}\right)^{\frac{1}{2\alpha}}$ *and* $H_2 = \frac{1}{2}\left(\frac{B\rho_2+AC}{\rho_1\rho_2}\nu^{1-\alpha}\right)^{\frac{1}{2}}$.

*Proof.* We need to analyze the last term from Lemma 11. Writing it down we obtain

$$\frac{B\rho_2+AC}{\rho_1\rho_2}L^2\sum_{t=0}^{T-1}\gamma_t^2\|g_t\|^2 = \frac{B\rho_2+AC}{\rho_1\rho_2}L^2\nu^{\alpha-1}\sum_{t=0}^{T-1}\frac{\|g_t\|^2}{\left(\sum\limits_{i=0}^{t-1}\|g_i\|^2\right)^{2\alpha}} \leq \frac{1}{1-2\alpha}\frac{B\rho_2+AC}{\rho_1\rho_2}L^2\nu^{\alpha-1}\left(\sum_{t=0}^{T-1}\|g_t\|^2\right)^{1-2\alpha}$$

$$= \frac{\nu^{\alpha-1}}{1-2\alpha}\frac{B\rho_2+AC}{\rho_1\rho_2}L^2\left(\frac{2-4\alpha}{1-\alpha}\left(\frac{B\rho_2+AC}{\rho_1\rho_2}\right)^{\frac{-1}{2}}\nu^{\frac{\alpha-1}{2}}L^{\frac{2\alpha-1}{1-\alpha}}\right)^{\frac{1-2\alpha}{1-\alpha}}$$

$$\cdot \left(\frac{1-\alpha}{2-4\alpha}\left(\frac{B\rho_2+AC}{\rho_1\rho_2}\right)^{\frac{1}{2}}\nu^{\frac{1-\alpha}{2}}L^{\frac{1-2\alpha}{1-\alpha}}\right)^{\frac{1-2\alpha}{1-\alpha}}\left(\sum_{t=0}^{T-1}\|g_t\|^2\right)^{1-2\alpha}$$

$$\leq H_1 + H_2\left(\sum_{t=0}^{T-1}\|g_t\|^2\right)^{1-\alpha},$$

where $H_1 = \frac{\alpha}{1-\alpha}\left(\frac{1}{1-2\alpha}\left(\frac{2-4\alpha}{1-\alpha}\right)^{\frac{1-2\alpha}{1-\alpha}}\right)^{\frac{1-\alpha}{\alpha}}\left(\frac{B\rho_2+AC}{\rho_1\rho_2}\nu^{1-\alpha}\right)^{\frac{1}{2\alpha}}L^{\frac{1}{\alpha}}$ and $H_2 =$

$\frac{1}{2}\left(\frac{B\rho_2+AC}{\rho_1\rho_2}\nu^{1-\alpha}\right)^{\frac{1}{2}}L$ $\qquad\qquad\qquad\qquad\qquad\qquad\qquad\qquad\qquad\qquad\qquad\qquad\square$

**Theorem 6.** *Let*

$$\gamma_t = \frac{1}{\nu^{\frac{1-\alpha}{2}}\left(\sum_{i=1}^{t}\|g_i\|^2\right)^{\alpha}}.$$

*Then we have*

$$\frac{1}{T}\sum_{t=0}^{T-1}\mathbb{E}\|\nabla f(x_t)\| = \mathcal{O}\left(\frac{V_0^{\frac{1}{2(1-\alpha)}}+L^{\frac{1}{2\alpha}}}{\sqrt{T}}\left(\nu^{\frac{\alpha-1}{4\alpha}}+\left(\frac{B\rho_2+AC}{\rho_1\rho_2}\right)^{\frac{1}{4\alpha}}\nu^{\frac{\alpha-1}{4\alpha}}+\left(\frac{B\rho_2+AC}{\rho_1\rho_2}\right)^{\frac{1}{4}}\nu^{\frac{\alpha-1}{4\alpha}}\right)\right).$$

*Proof.* Start from the decomposition
$$\mathbb{E}\|\nabla f(x_t)\| \leq \mathbb{E}\|g_t\| + \mathbb{E}\|g_t - \nabla f(x_t)\|.$$
Averaging over $T$ iterates gives
$$\frac{1}{T}\sum_{t=0}^{T-1}\mathbb{E}\|\nabla f(x_t)\| \leq \frac{1}{T}\sum_{t=0}^{T-1}\mathbb{E}\|g_t\| + \frac{1}{T}\sum_{t=0}^{T-1}\mathbb{E}\|g_t - \nabla f(x_t)\|.$$
Bound the first term using Lemma 10. Lemma 10 gives
$$\mathbb{E}\left(\frac{1}{\sqrt{\nu}}\sum_{t=0}^{T-1}\|g_t\|^2\right)^{1-\alpha} \leq 6V_0 + 3G_1\left(\frac{L}{\sqrt{\nu}}\right)^{\frac{1-\alpha}{\alpha}} + 3G_2\left(\frac{B\rho_2+AC}{\rho_1\rho_2\nu}\right)^{\frac{1-\alpha}{2\alpha}}.$$
Taking both sides to the power $\frac{1}{2(1-\alpha)}$ and using Jensen's inequality for the average over $T$, we obtain

$$\frac{1}{T}\sum_{t=0}^{T-1}\mathbb{E}\|g_t\| \leq \left(\mathbb{E}\left(\frac{1}{\sqrt{\nu}}\sum_{t=0}^{T-1}\|g_t\|^2\right)^{1-\alpha}\right)^{\frac{1}{2(1-\alpha)}}\cdot\nu^{1/4}$$

$$\leq \nu^{1/4}\left[(6V_0)^{\frac{1}{2(1-\alpha)}}+\left(\left(\frac{L}{\sqrt{\nu}}\right)^{\frac{1-\alpha}{\alpha}}\right)^{\frac{1}{2(1-\alpha)}}+\left(\left(\frac{B\rho_2+AC}{\rho_1\rho_2\nu}\right)^{\frac{1-\alpha}{2\alpha}}\right)^{\frac{1}{2(1-\alpha)}}\right].$$

Explicitly, this gives

$$\frac{1}{T}\sum_{t=0}^{T-1}\mathbb{E}\|g_t\| = \mathcal{O}\left(\left(V_0^{\frac{1}{2(1-\alpha)}} + L^{\frac{1}{2\alpha}}\right)\left(\nu^{1/4} + \left(\frac{1}{\sqrt{\nu}}\right)^{\frac{1}{2\alpha}}\nu^{1/4} + \left(\frac{B\rho_2 + AC}{\rho_1\rho_2\nu}\right)^{\frac{1}{4\alpha}}\nu^{1/4}\right)\right).$$

Bound the second term using Lemma 11. Lemma 11 implies

$$\frac{1}{T}\sum_{t=0}^{T-1}\mathbb{E}\|g_t - \nabla f(x_t)\| \le \sqrt{\frac{1}{T}\sum_{t=0}^{T-1}\mathbb{E}\|g_t - \nabla f(x_t)\|^2} = \mathcal{O}\left(\frac{1}{\sqrt{T}}\left(\|g_0 - \nabla f(x_0)\| + \sigma_0 + H_1^{1/2} + H_2^{1/2}\mathbb{E}\left(\sum_{t=0}^{T-1}\|g_t\|^2\right)^{\frac{1-\alpha}{2}}\right)\right)$$

where $H_1, H_2$ are defined in Lemma 11.

Combining these bounds we obtain the needed. $\square$

**Corollary 7.** *Let*

$$\nu = \max\left\{\frac{B\rho_2 + AC}{\rho_1\rho_2}, 1\right\}.$$

*Then we have*

$$\frac{1}{T}\sum_{t=0}^{T-1}\mathbb{E}\|\nabla f(x_t)\| = \mathcal{O}\left(\frac{\max\left\{\left(\frac{B\rho_2 + AC}{\rho_1\rho_2}\right)^{1/4}, 1\right\}}{\sqrt{T}}\right).$$

*Proof.* From the theorem, we have the bound

$$\frac{1}{T}\sum_{t=0}^{T-1}\mathbb{E}\|\nabla f(x_t)\| = \mathcal{O}\left(\frac{1}{\sqrt{T}}\left(\nu^{\frac{\alpha-1}{4\alpha}} + \left(\frac{B\rho_2 + AC}{\rho_1\rho_2}\right)^{\frac{1}{4\alpha}}\nu^{\frac{\alpha-1}{4\alpha}} + \left(\frac{B\rho_2 + AC}{\rho_1\rho_2}\right)^{\frac{1}{4}}\nu^{\frac{\alpha-1}{4\alpha}}\right)\right).$$

**Case 1:** If $\frac{B\rho_2 + AC}{\rho_1\rho_2} \le 1$, then $\nu = 1$, and both terms become at most order 1. So the bound reduces to

$$\frac{1}{T}\sum_{t=0}^{T-1}\mathbb{E}\|\nabla f(x_t)\| = \mathcal{O}\left(\frac{1}{\sqrt{T}}\left(1 + \left(\frac{B\rho_2 + AC}{\rho_1\rho_2}\right)^{\frac{1}{4\alpha}} + \left(\frac{B\rho_2 + AC}{\rho_1\rho_2}\right)^{\frac{1}{4}}\right)\right) = \mathcal{O}\left(\frac{1}{\sqrt{T}}\right).$$

**Case 2:** If $\frac{B\rho_2 + AC}{\rho_1\rho_2} > 1$, then $\nu = \frac{B\rho_2 + AC}{\rho_1\rho_2}$. In this case, bound reduces to

$$\frac{1}{T}\sum_{t=0}^{T-1}\mathbb{E}\|\nabla f(x_t)\| = \mathcal{O}\left(\frac{1}{\sqrt{T}}\left(\left(\frac{B\rho_2 + AC}{\rho_1\rho_2}\right)^{\frac{\alpha-1}{4\alpha}} + \left(\frac{B\rho_2 + AC}{\rho_1\rho_2}\right)^{\frac{2\alpha-1}{4\alpha}} + \left(\frac{B\rho_2 + AC}{\rho_1\rho_2}\right)^{\frac{1}{4}}\right)\right).$$

Having bounds on $\alpha$, we get $\alpha - 1 \le 2\alpha - 1 \le -\frac{1}{3} < 0$. Therefore, with $\frac{B\rho_2 + AC}{\rho_1\rho_2} > 1$ the most impactful term is $\left(\frac{B\rho_2 + AC}{\rho_1\rho_2}\right)^{\frac{1}{4}}$

Combining both cases, we can write the bound compactly using a maximum:

$$\frac{1}{T}\sum_{t=0}^{T-1}\mathbb{E}\|\nabla f(x_t)\| = \mathcal{O}\left(\frac{\max\left\{\left(\frac{B\rho_2 + AC}{\rho_1\rho_2}\right)^{1/4}, 1\right\}}{\sqrt{T}}\right),$$

which proves the corollary. $\square$

## C  FAMILY OF ESTIMATORS

In this section we provide proofs that mentioned estimators satisfies Assumption 1. First of all, we establish the technical lemmas.

**Lemma 13** (Young's Inequality). *Let $x, y \in \mathbb{R}^d$, then $\forall \alpha > 0$ we have*

$$\langle x, y \rangle \leq \frac{\alpha}{2}\|x\|^2 + \frac{2}{\alpha}\|y\|^2 \tag{33}$$

*and*

$$\|x + y\|^2 \leq (1 + \alpha)\|x\|^2 + \left(1 + \frac{1}{\alpha}\right)\|y\|^2. \tag{34}$$

**Lemma 14** (Lemma A.1 from (Lei et al., 2017)). *Let $x_1, \ldots, x_N \in \mathbb{R}^d$ be arbitrary vectors with*

$$\sum_{i=1}^{N} x_i = 0.$$

*Let $S$ be a uniform subset of $\{1, \ldots, N\}$ with size $b$. Then*

$$\mathbb{E}\left\|\frac{1}{b}\sum_{i \in S} x_i\right\|^2 \leq \frac{1}{bN}\sum_{i=1}^{N}\|x_i\|^2 \tag{35}$$

### C.1  L-SVRG

**Lemma 15.** *L-SVRG satisfies Assumption 1 with:*

$$\rho_1 = 1, \ A = \frac{2}{b}, \ B = \frac{2}{b},$$

$$\sigma_t^2 = \frac{1}{n}\sum_{i=1}^{n}\|\nabla f_i(w^{t+1}) - \nabla f_i(x^t)\|^2, \ \rho_2 = \frac{p}{2}, \ C = 1 + \frac{2}{p}.$$

*Proof.* We bound the difference between the gradient estimator and exact gradient

$$
\begin{aligned}
\mathbb{E}_t\left[\|g^t - \nabla f(x^t)\|^2\right] &= \mathbb{E}_t\left[\left\|\frac{1}{b}\sum_{i \in S_t}\left[\nabla f_i(x^t) - \nabla f_i(w^t)\right] + \frac{1}{n}\sum_{j=1}^{n}\nabla f_j(w^t) - \nabla f(x^t)\right\|^2\right] \\
&= \mathbb{E}_t\left[\left\|\frac{1}{b}\left(\sum_{i \in S_t}\left[\nabla f_i(x^t) - \nabla f_i(w^t)\right] - \left(\frac{1}{n}\sum_{j=1}^{n}\left[\nabla f_j(x^t) - y_j^t\right]\right)\right)\right\|^2\right] \\
&\overset{(14)}{\leq} \frac{1}{bn}\sum_{j=1}^{n}\left\|\nabla f_j(x^t) - \nabla f_j(w^t) - \left(\frac{1}{n}\sum_{i=1}^{n}\left[\nabla f_i(x^t) - \nabla f_i(w^t)\right]\right)\right\|^2 \\
&\leq \frac{1}{bn}\sum_{j=1}^{n}\left\|\nabla f_j(x^t) - \nabla f_j(w^t)\right\|^2 \\
&\leq \frac{2}{b}\sum_{i=1}^{n}\|\nabla f_i(w^t) - \nabla f_i(x^{t-1})\|^2 + \frac{2}{b}\sum_{i=1}^{n}\|f_i(x^t) - f_i(x^{t-1})\|^2 \\
&\leq \frac{2}{b}\sum_{i=1}^{n}\|\nabla f_i(w^t) - \nabla f_i(x^{t-1})\|^2 + \frac{2L^2}{b}\|x^t - x^{t-1}\|^2
\end{aligned}
$$

The second inequality holds, since $\frac{1}{n}\sum_{i=1}^{n}$ can be described, as an expected value. And $\mathbb{E}\|x - \mathbb{E}x\|^2 \le \mathbb{E}\|x\|^2$. Then we need to bound the first term:

$$
\begin{aligned}
\mathbb{E}_t \frac{1}{n}\sum_{i=1}^{n}\|\nabla f_i(w^{t+1}) - \nabla f_i(x^t)\|^2 \;=\;& (1-p)\frac{1}{n}\sum_{i=1}^{n}\|\nabla f_i(w^t) - \nabla f_i(x^t)\|^2 \\
\le\;& (1-p)\left(1 + \frac{p}{2}\right)\frac{1}{n}\sum_{i=1}^{n}\|\nabla f_i(w^t) - \nabla f_i(x^{t-1})\|^2 \\
+\;& (1-p)\left(1 + \frac{2}{p}\right)\frac{1}{n}\sum_{i=1}^{n}\|\nabla f_i(x^t) - \nabla f_i(x^{t-1})\|^2 \\
\le\;& \left(1 - \frac{p}{2}\right)\sum_{i=1}^{n}\|\nabla f_i(w^t) - \nabla f_i(x^{t-1})\|^2 \\
+\;& \left(1 + \frac{2}{p}\right)\frac{1}{n}\sum_{i=1}^{n}\|\nabla f_i(x^t) - \nabla f_i(x^{t-1})\|^2 \\
\le\;& \left(1 - \frac{p}{2}\right)\sum_{i=1}^{n}\|\nabla f_i(w^t) - \nabla f_i(x^{t-1})\|^2 \\
+\;& \left(1 + \frac{2}{p}\right) L^2\|x^t - x^{t-1}\|^2.
\end{aligned}
$$

$\square$

## C.2   SAGA

**Lemma 16.** *SAGA satisfies Assumption 1 with:*

$$
\rho_1 = 1,\; A = \frac{1}{b}\left(1 + \frac{b}{2n}\right),\; B = \frac{1}{b}\left(1 + \frac{2n}{b}\right),
$$

$$
\sigma_t^2 = \frac{1}{n}\sum_{j=1}^{n}\|\nabla f_j(x^t) - y_j^{t+1}\|^2,\; \rho_2 = \frac{b}{2n},\; C = \frac{2n}{b}.
$$

*Proof.* We bound the difference between estimator and exact gradient:

$$
\begin{aligned}
\mathbb{E}_t\left[\|g^t - \nabla f(x^t)\|^2\right] \;=\;& \mathbb{E}_t\left[\left\|\frac{1}{b}\sum_{i \in S_t}\left[\nabla f_i(x^t) - y_i^t\right] + \frac{1}{n}\sum_{j=1}^{n}y_j^t - \nabla f(x^t)\right\|^2\right] \\
=\;& \mathbb{E}_t\left[\left\|\frac{1}{b}\left(\sum_{i \in S_t}\left[\nabla f_i(x^t) - y_i^t\right] - \left(\frac{1}{n}\sum_{j=1}^{n}\left[\nabla f_j(x^t) - y_j^t\right]\right)\right)\right\|^2\right] \\
\overset{(14)}{\le}\;& \frac{1}{bn}\sum_{j=1}^{n}\left\|\nabla f_j(x^t) - y_j^t - \left(\frac{1}{n}\sum_{i=1}^{n}\left[\nabla f_i(x^t) - y_i^t\right]\right)\right\|^2 \\
\le\;& \frac{1}{bn}\sum_{j=1}^{n}\left\|\nabla f_j(x^t) - y_j^t\right\|^2 \\
\le\;& \frac{1}{bn}(1+\alpha)\sum_{j=1}^{n}\|\nabla f_j(x^t) - \nabla f_j(x^{t-1})\|^2 + \frac{1}{bn}\left(1 + \frac{1}{\alpha}\right)\sum_{j=1}^{n}\|\nabla f_j(x^{t-1}) - y_j^t\|^2 \\
\le\;& \frac{L^2}{b}(1+\alpha)\|x^t - x^{t-1}\|^2 + \frac{1}{b}\left(1 + \frac{1}{\alpha}\right)\sigma_{t-1}^2
\end{aligned}
$$

for $\forall \alpha > 0$ (in particular, we can put $\alpha = \frac{2n}{b}$ to obtain the needed estimates). The second inequality holds, since $\frac{1}{n}\sum_{i=1}^{n}$ can be described, as an expected value. And $\mathbb{E}\|x - \mathbb{E}x\|^2 \le \mathbb{E}\|x\|^2$. Then we

need to bound the second term:

$$
\begin{aligned}
\mathbb{E}_t[\sigma_t^2] &= \mathbb{E}_t\left[\frac{1}{n}\sum_{j=1}^{n}\|\nabla f_j(x^t) - y_j^{t+1}\|^2\right] = \left(1 - \frac{b}{n}\right)\frac{1}{n}\sum_{j=1}^{n}\|\nabla f_j(x^t) - y_j^t\|^2 \\
&= \left(1 - \frac{b}{n}\right)\frac{1}{n}\sum_{j=1}^{n}\|\nabla f_j(x^t) - \nabla f_j(x^{t-1}) + \nabla f_j(x^{t-1}) - y_j^{t-1}\|^2 \\
&\leq \left(1 - \frac{b}{n}\right)(1+\beta)\frac{1}{n}\sum_{j=1}^{n}\|\nabla f_j(x^{t-1}) - y_j^{t-1}\|^2 + \left(1 - \frac{b}{n}\right)\left(1 + \frac{1}{\beta}\right)L^2\|x^t - x^{t-1}\|^2.
\end{aligned}
$$

With $\beta = \frac{b}{2n}$ we have:

$$
\mathbb{E}_t[\sigma_t^2] \leq \left(1 - \frac{b}{2n}\right)\sigma_{t-1}^2 + \frac{2n}{b}L^2\|x^t - x_{t-1}\|^2.
$$

$\square$

## C.3   PAGE

**Lemma 17.** *PAGE satisfies Assumption 1 with:*

$$
\rho_1 = p,\ A = 0,\ B = \frac{1-p}{b},\ C = 0,
$$
$$
\sigma_t = 0,\ \rho_2 = 1,\ E = 0.
$$

*Proof.* Using Lemma 3 from (Li et al., 2021a) we can obtain:

$$
\mathbb{E}_t\left[\|\nabla f(x^t) - g^t\|^2\right] \leq (1-p)\|\nabla f(x^{t-1}) - g^{t-1}\|^2 + \frac{1-p}{b}L^2\|x^t - x^{t-1}\|^2.
$$

$\square$

## C.4   ZEROSARAH

**Lemma 18.** *ZeroSARAH satisfies Assumption 1 with:*

$$
\rho_1 = \frac{b}{2n},\ A = \frac{b}{2n^2},\ B = \frac{2}{b},
$$
$$
\sigma_t^2 = \frac{1}{n}\sum_{j=1}^{n}\mathbb{E}[\|\nabla f_j(x^t) - y_j^{t+1}\|^2],\ \rho_2 = \frac{b}{2n},\ C = \frac{2n}{b}.
$$

*Proof.* Using Lemma 2 from (Li et al., 2021b) we can obtain:

$$
\begin{aligned}
\mathbb{E}_t\left[\|\nabla f(x^t) - g^t\|^2\right] &\leq (1-\lambda)^2\|\nabla f(x^{t-1}) - g^{t-1}\|^2 + \frac{2\lambda^2}{b}\frac{1}{n}\sum_{j=1}^{n}\|\nabla f_j(x^{t-1}) - y_j^t\|^2 \\
&\quad + \frac{2L}{b}\|x^t - x^{t-1}\|^2 \\
&\leq (1-\lambda)^2\|\nabla f(x^t) - g^t\|^2 + \frac{2\lambda^2}{b}\frac{1}{n}\sum_{j=1}^{n}\|\nabla f_j(x^{t-1}) - y_j^t\|^2 + \frac{2L}{b}.
\end{aligned}
$$

Additionally Lemma 3 from (Li et al., 2021b) with $\beta_t = \frac{b}{2n}$ gives us:

$$
\frac{1}{n}\sum_{j=1}^{n}\|\nabla f_j(x^t) - y_j^{t+1}\|^2 \leq \left(1 - \frac{b}{2n}\right)\frac{1}{n}\sum_{j=1}^{n}\|\nabla f_j(x^{t-1}) - y_j^t\|^2 + \frac{2nL^2}{b}\|x^t - x^{t-1}\|^2.
$$

With $\lambda = \frac{b}{2n}$ we have:

$$
\mathbb{E}_t\left[\|\nabla f(x^t) - g^t\|^2\right] \leq \left(1 - \frac{b}{2n}\right)\|\nabla f(x^{t-1}) - g^t\|^2 + \frac{b}{2n^2}\frac{1}{n}\sum_{j=1}^{n}\|\nabla f_j(x^{t-1}) - y_j^t\|^2 + \frac{2L}{b}\|x^t - x^{t-1}\|^2.
$$

$\square$

## C.5  EF21

**Lemma 19.** *EF21 satisfies Assumption 1 with:*
$$\rho_1 = 1, \; A = 1, \; B = 0,$$
$$\sigma_t^2 = \frac{1}{n}\sum_{i=1}^{n}\|g_i^t - \nabla f_i(x^t)\|^2, \; \rho_2 = \frac{\delta+1}{2\delta^2}, \; E = 2\delta.$$

*Proof.* First, let us notice:
$$\mathbb{E}_t\left[\|g^t - \nabla f(x^t)\|^2\right] = \mathbb{E}_t\left[\left\|\frac{1}{n}\sum_{i=1}^{n}\left(g_i^t - \nabla f_i(x^t)\right)\right\|^2\right] \leq \frac{1}{n}\sum_{i=1}^{n}\mathbb{E}_t\left[\|g_i^t - \nabla f_i(x^t)\|^2\right].$$

Similar to the Proof of Theorem 1 from (Richtárik et al., 2021), we can derive:

$$
\begin{aligned}
\frac{1}{n}\sum_{i=1}^{n}\mathbb{E}_t\left[\|g_i^t - \nabla f_i(x^t)\|^2\right] &= \frac{1}{n}\sum_{i=1}^{n}\mathbb{E}_t\left[\|g_i^{t-1} + \mathcal{C}(\nabla f_i(x^t) - g_i^{t-1}) - \nabla f_i(x^t)\|^2\right] \\
&\leq \left(1 - \frac{1}{\delta}\right)\frac{1}{n}\sum_{i=1}^{n}\|g_i^{t-1} - \nabla f_i(x^t)\|^2 \\
&\leq \left(1 - \frac{1}{\delta}\right)(1+\alpha)\frac{1}{n}\sum_{i=1}^{n}\|g_i^{t-1} - \nabla f_i(x^{t-1})\|^2 \\
&\quad + \left(1 - \frac{1}{\delta}\right)\left(1 + \frac{1}{\alpha}\right)L^2\|x^t - x^{t-1}\|^2.
\end{aligned}
$$

for any $\alpha > 0$. Choose $\alpha = \frac{1}{2\delta}$, hence
$$\frac{1}{n}\sum_{i=1}^{n}\mathbb{E}_t\left[\|g_i^t - \nabla f_i(x^t)\|^2\right] \leq \left(1 - \frac{\delta+1}{2\delta^2}\right)\frac{1}{n}\sum_{i=1}^{n}\|g_i^{t-1} - \nabla f_i(x^{t-1})\|^2 + 2\delta L^2\|x^t - x_{t-1}\|^2.$$
$\square$

## C.6  DIANA

One of the first methods to incorporate the error compensating technique with unbiased compressors, was the DIANA method (Mishchenko et al., 2019).
$$
\begin{aligned}
\Delta_i^t &= \mathcal{Q}\left(\nabla f_i(x^t) - h_i^t\right), \; h_i^{t+1} = h_i^t + \frac{1}{\omega+1}\Delta_i^t, \\
h^{t+1} &= h^t + \frac{1}{\omega+1}\cdot\frac{1}{n}\sum_{i=1}^{n}\Delta_i^t, \\
g^t &= h^{t+1} + \frac{1}{n}\sum_{i=1}^{n}\Delta_i^t.
\end{aligned}
\tag{36}
$$
As EF21, this algorithm also compresses the differences, but due to the unbiased nature it needs an additional "memory" sequence $h_i^t$ at each client.

**Lemma 20.** *DIANA satisfies Assumption 1 with:*
$$\rho_1 = 1, \; A = \frac{\omega}{n^2}, \; B = \frac{2\omega(\omega+1)}{n},$$
$$\sigma_t^2 = \sum_{i=1}^{n}\|\nabla f_i(x^t) - h_i^t\|^2, \; \rho_2 = \frac{1}{2(1+\omega)}, \; C = 2(\omega+1)n.$$

*Proof.* Deriving inequalities from the proof of Theorem 7 from (Li & Richtárik, 2020), we get
$$
\begin{aligned}
\mathbb{E}_t\left[\|g^t - \nabla f(x^t)\|^2\right] &\leq \frac{\omega}{n^2}\mathbb{E}_t\left[\sum_{i=1}^{n}\|\nabla f_i(x^t) - h_i^t\|^2\right] \\
\mathbb{E}_t\left[\sum_{i=1}^{n}\|\nabla f_i(x^t) - h_i^t\|^2\right] &\leq \left(1 - 2\alpha + \frac{(1-\alpha)^2}{\beta} + \alpha^2(1+\omega)\right)\sum_{i=1}^{n}\mathbb{E}_t\left[\|\nabla f_i(x^{t-1}) - h_i^{t-1}\|^2\right] \\
&\quad + (1+\beta)\sum_{i=1}^{n}\mathbb{E}_t\left[\|\nabla f_i(x^t) - \nabla f_i(x^{t-1})\|^2\right]
\end{aligned}
$$

for $\forall \beta > 0$. Choose $\beta = \frac{2\omega^2}{1+\omega}$, then

$$
\begin{aligned}
\mathbb{E}_t \left[ \sum_{i=1}^n \| \nabla f_i(x^t) - h_i^t \|^2 \right] &\leq \frac{\omega + \frac{1}{2}}{\omega + 1} \sum_{i=1}^n \mathbb{E}_t \left[ \| \nabla f_i(x^{t-1}) - h_i^{t-1} \|^2 \right] \\
&+ \frac{2\omega^2 + \omega + 1}{\omega + 1} n L^2 \| x^t - x^{t-1} \|^2, \\
&\leq \left( 1 - \frac{1}{2(1+\omega)} \right) \sum_{i=1}^n \mathbb{E}_t \left[ \| \nabla f_i(x^{t-1}) - h_i^{t-1} \|^2 \right] \\
&+ 2(\omega+1) n L^2 \| x^t - x^{t-1} \|^2. \\
\mathbb{E}_t \left[ \| g^t - \nabla f(x^t) \|^2 \right] &\leq \frac{\omega}{n^2} \sum_{i=1}^n \mathbb{E}_t \left[ \| \nabla f_i(x^{t-1}) - h_i^{t-1} \|^2 \right] + 2 \frac{\omega}{n} (\omega+1) L^2 \| x^t - x^{t-1} \|^2.
\end{aligned}
$$

$\square$

**Corollary 8.** *In the non-convex case choosing step sizes as*

$$
\gamma \lesssim \left( L \left[ 1 + \frac{\omega^{3/2}}{\sqrt{n}} \right] \right)^{-1} \quad \text{results in} \quad \mathbb{E} \| \nabla f(x^\tau) \|^2 = \mathcal{O} \left( \frac{1}{T} \left( 1 + \frac{\omega^{3/2}}{\sqrt{n}} \right) \right). \tag{37}
$$

*Taking adaptive step sizes as*

$$
\gamma_t = \frac{1}{\left( \max \left\{ \frac{\omega^{3/2}}{\sqrt{n}}; 1 \right\} \right)^{1-\alpha} \left( \sum_{i=0}^{t-1} \| g^i \|^2 \right)^{\alpha}} \quad \text{results in} \quad \mathbb{E} \| \nabla f(x^\tau) \|^2 = \mathcal{O} \left( \frac{\max \left\{ \frac{\omega^{3/2}}{\sqrt{n}}; 1 \right\}}{T} \right). \tag{38}
$$

*Here $\tau$ is chosen uniformly over $0, \ldots, T-1$.*

### C.7 DASHA

**Lemma 21.** *DASHA satisfies Assumption 1 with:*

$$
\rho_1 = \frac{1}{2\omega + 1}, \; A = \frac{2\omega}{(2\omega+1)^2 n}, \; B = \frac{2\omega}{n},
$$

$$
\sigma_t^2 = \frac{1}{n} \sum_{i=1}^n \| g_i^t - \nabla f_i(x^t) \|^2, \; \rho_2 = \frac{1}{2\omega + 1}, \; C = 2\omega.
$$

*Proof.* From (Tyurin & Richtárik, 2022) we get

$$
\begin{aligned}
\mathbb{E}_t \| g^t - \nabla f(x^t) \|^2 &\leq \left( 1 - \frac{1}{2\omega + 1} \right)^2 \| g^{t-1} - \nabla f(x^{t-1}) \|^2 \\
&+ \frac{2\omega}{(2\omega+1)^2 n^2} \sum_{i=1}^n \| g_i^{t-1} - \nabla f_i(x^{t-1}) \|^2 \frac{2\omega L^2}{n} \| x^t - x^{t-1} \|^2 \\
&\leq \left( 1 - \frac{1}{2\omega + 1} \right) \| g^{t-1} - \nabla f(x^{t-1}) \|^2 \\
&+ \frac{2\omega}{(2\omega+1)^2 n^2} \sum_{i=1}^n \| g_i^{t-1} - \nabla f_i(x^{t-1}) \|^2 + \frac{2\omega L^2}{n} \| x^t - x^{t-1} \|^2.
\end{aligned}
$$

For the second term we also inherit the following bound:

$$
\begin{aligned}
\mathbb{E}_t \frac{1}{n} \sum_{i=1}^n \| g_i^t - \nabla f_i(x^t) \|^2 &\leq \left( \frac{2\omega}{(2\omega+1)^2} + \left( 1 - \frac{1}{2\omega+1} \right)^2 \right) \frac{1}{n} \sum_{i=1}^n \| g_i^{t-1} - \nabla f_i(x^{t-1}) \|^2 \\
&+ 2\omega L^2 \| x^t - x^{t-1} \|^2 \\
&\leq \left( 1 - \frac{1}{2\omega+1} \right) \frac{1}{n} \sum_{i=1}^n \| g_i^{t-1} - \nabla f_i(x^{t-1}) \|^2 + 2\omega L^2 \| x^t - x^{t-1} \|^2
\end{aligned}
$$

$\square$

## C.8 SEGA

Previous approaches reduce the computational costs by either selecting random batches ore compressing messages. Another option is to compute partial derivatives, instead of full gradients. This may be beneficial, if there is a clear analytical expression for them. Also, partial derivatives may be approximated via zero order methods, which makes these methods more effective.

**SEGA.** As in DIANA, storing an additional "memory" sequence may enhance convergence. This idea was firstly implemented in (Hanzely et al., 2018), where a bit more general setting was considered. We use a simplified version, where the gradient estimator $g^t$ is updated as following:

$$
\begin{aligned}
h^t &= h^{t-1} + e_{i_t}\left(\nabla_{i_t} f(x^{t-1}) - h^{t-1}_{i_t}\right) \\
g^t &= d\left(\nabla_{i_t} f(x^t) - h^t_{i_t}\right) e_{i_t} + h^t,
\end{aligned} \tag{39}
$$

where coordinate $i_t$ is chosen independently and uniformly.

**Lemma 22.** *SEGA satisfies Assumption 1 with:*

$$
\rho_1 = 1, \ A = d, \ B = d^2
$$

$$
\sigma_t^2 = \|h^{t+1} - \nabla f(x^t)\|^2, \ \rho_2 = \frac{1}{2d}, \ C = 3d.
$$

*Proof.* We first bound the difference between estimator and exact gradient:

$$
\begin{aligned}
\mathbb{E}_t\left[\|g^t - \nabla f(x^t)\|^2\right] &= \mathbb{E}_t\left[\|de_{i_t}e_{i_t}^T(\nabla f(x^t) - h^t) + h^t - \nabla f(x^t)\|^2\right] \\
&= \mathbb{E}_t\left[\|(I - de_{i_t}e_{i_t}^T)(h^t - \nabla f(x^t))\|^2\right] \\
&= \mathbb{E}_t\left[(h^t - \nabla f(x^t))^T(I - de_{i_t}e_{i_t}^T)^T(I - de_{i_t}e_{i_t}^T)(h^t - \nabla f(x^t))\right] \\
&= (h^t - \nabla f(x^t))^T\mathbb{E}_t\left[I - 2de_{i_t}e_{i_t}^T + d^2 e_{i_t}e_{i_t}^T\right](h^t - \nabla f(x^t)) \\
&= (h^t - \nabla f(x^t))^T\left[I - 2\cdot I + d\cdot I\right](h^t - \nabla f(x^t)) \\
&= (d-1)\|h^t - \nabla f(x^t)\|^2 \\
&\leq (d-1)(1+\alpha)\|h^t - \nabla f(x^{t-1})\|^2 \\
&+ (d-1)\left(1 + \frac{1}{\alpha}\right)L^2\|x^t - x^{t-1}\|^2.
\end{aligned}
$$

Then,

$$
\begin{aligned}
\mathbb{E}_t\left[\|h^{t+1} - \nabla f(x^t)\|^2\right] &= \mathbb{E}_t\left[\|h^t + e_{i_t}e_{i_t}^T(\nabla f(x^t) - h^t) - \nabla f(x^t)\|^2\right] \\
&= \mathbb{E}_t\left[\|(I - e_{i_t}e_{i_t}^T)(h^t - \nabla f(x^t))\|^2\right] \\
&= \left(1 - \frac{1}{d}\right)\|h^t - \nabla f(x^t)\|^2 \\
&\leq \left(1 - \frac{1}{d}\right)(1+\beta)\|h^t - \nabla f(x^{t-1})\|^2 \\
&+ \left(1 - \frac{1}{d}\right)\left(1 + \frac{1}{\beta}\right)L^2\|x^t - x^{t-1}\|^2.
\end{aligned}
$$

If $\beta = \frac{1}{2d}$ then $(1 - \frac{1}{d})(1 + \frac{1}{2d}) \leq 1 - \frac{1}{2d}$ and $(1 - \frac{1}{d})(1 + 2d) \leq 2d$, then as $d \geq 1$:

$$
\mathbb{E}_t\left[\|h^{t+1} - \nabla f(x^t)\|^2\right] \leq \left(1 - \frac{1}{2d}\right)\|h^t - \nabla f(x^{t-1})\|^2 + 3dL^2\|x^t - x^{t-1}\|^2.
$$

Taking $\alpha = \frac{1}{d}$, we obtain the needed constants. $\qquad\square$

**Corollary 9.** *In the non-convex case choosing step sizes as*

$$
\gamma \lesssim \left(L\left[1 + d\sqrt{d}\right]\right)^{-1} \quad \textit{results in} \quad \mathbb{E}\|\nabla f(x^\tau)\|^2 = \mathcal{O}\left(\frac{1}{T}\left(1 + d\sqrt{d}\right)\right). \tag{40}
$$

*Taking adaptive step sizes as*

$$
\gamma_t = \frac{1}{d^{\frac{3-3\alpha}{2}}\left(\sum_{i=0}^{t-1}\|g^i\|^2\right)^\alpha} \quad \textit{results in} \quad \mathbb{E}\|\nabla f(x^\tau)\|^2 = \mathcal{O}\left(\frac{d^{3/4}}{\sqrt{T}}\right). \tag{41}
$$

*Here $\tau$ is chosen uniformly over $0, \ldots, T-1$.*

### C.9 JAGUAR

**JAGUAR.** Besides SEGA, we consider JAGUAR (Veprikov et al., 2024) method, as its gradient estimation is biased and can not be described in previous unified analyses:

$$g^t = g^{t-1} + \sum_{i \in S_t} e_i \left( \nabla_i f(x^t) - g^{t-1} \right), \tag{42}$$

where mini-batches $S_t$ of size $b$ are generated independently and uniformly.

**Lemma 23.** *JAGUAR satisfies Assumption 1 with:*

$$\rho_1 = \frac{1}{2d}, A = 0, \ B = 3d,$$
$$\sigma_t^2 = 0, \ \rho_2 = 1, C = 0.$$

*Proof.* We first bound the difference between estimator and exact gradient:

$$
\begin{aligned}
\mathbb{E}_t \left[ \|g^t - \nabla f(x^t)\|^2 \right] &= \mathbb{E}_t \left[ \|e_{i_t} e_{i_t}^T (\nabla f(x^{t-1}) - g^{t-1}) + g^{t-1} - \nabla f(x^t)\|^2 \right] \\
&= \mathbb{E}_t \left[ \|e_{i_t} e_{i_t}^T (\nabla f(x^{t-1}) - g^{t-1}) + g^{t-1} - \nabla f(x^t) + \nabla f(x^{t-1}) - \nabla f(x^{t-1})\|^2 \right] \\
&= \mathbb{E}_t \left[ \|(I - e_{i_t} e_{i_t}^T)(\nabla f(x^{t-1}) - g^{t-1}) + \nabla f(x^{t-1}) - \nabla f(x^t)\|^2 \right] \\
&\leq (1+\beta)\mathbb{E}_t \left[ \|(I - e_{i_t} e_{i_t}^T)(g^{t-1} - \nabla f(x^{t-1}))\|^2 \right] + \left(1 + \frac{1}{\beta}\right) L^2 \|x^t - x^{t-1}\|^2 \\
&= (1+\beta)\left(1 - \frac{1}{d}\right) \|g^{t-1} - \nabla f(x^{t-1})\|^2 + \left(1 + \frac{1}{\beta}\right) L^2 \|x^t - x^{t-1}\|^2.
\end{aligned}
$$

If $\beta = \frac{1}{2d}$ then $(1 - \frac{1}{d})(1 + \frac{1}{2d}) \leq 1 - \frac{1}{2d}$ and then as $d \geq 1$ :

$$\mathbb{E}_t \left[ \|g^t - \nabla f(x^t)\|^2 \right] \leq \left(1 - \frac{1}{2d}\right) \|g^{t-1} - \nabla f(x^{t-1})\|^2 + 3dL^2 \|x^t - x^{t-1}\|^2.$$

This finishes the proof. $\qquad\square$

**Corollary 10.** *In the non-convex case choosing step sizes as*

$$\gamma \lesssim \left( L \left[ 1 + \frac{d}{b} \right] \right)^{-1} \quad \text{results in} \quad \mathbb{E} \|\nabla f(x^\tau)\|^2 = \mathcal{O}\left( \frac{1}{T}\left(1 + \frac{d}{b}\right) \right). \tag{43}$$

*Taking adaptive step sizes as*

$$\gamma_t = \frac{b^{1-\alpha}}{d^{1-\alpha} \left( \sum_{i=0}^{t-1} \|g^i\|^2 \right)^\alpha} \quad \text{results in} \quad \mathbb{E} \|\nabla f(x^\tau)\|^2 = \mathcal{O}\left( \frac{d^{1/2}}{T} \right). \tag{44}$$

*Here $\tau$ is chosen uniformly over $0, \dots, T-1$.*

We have shown, that various distributed optimization algorithms can be described not only with proposed unified scheme, but also be implemented with adaptive step sizes. To our knowledge, these are first distributed adaptive algorithms, which are, moreover, parameter-free. Adaptive algorithms' variations have the same asymptotic $\mathcal{O}\left(1/\sqrt{T}\right)$, as non-adaptive. It is optimal in non-convex scenario and cannot be improved.

# D    ADDITIONAL NUMERICAL EXPERIMENTS

## D.1    $\alpha$ ABLATION STUDY

Firstly, we analyze the different choices of parameter $\alpha \in (0, \frac{1}{3})$. We take one method per considered class: SAGA for finite sum problem, EF21 for distributed optimization and JAGUAR from coordinate-based methods.

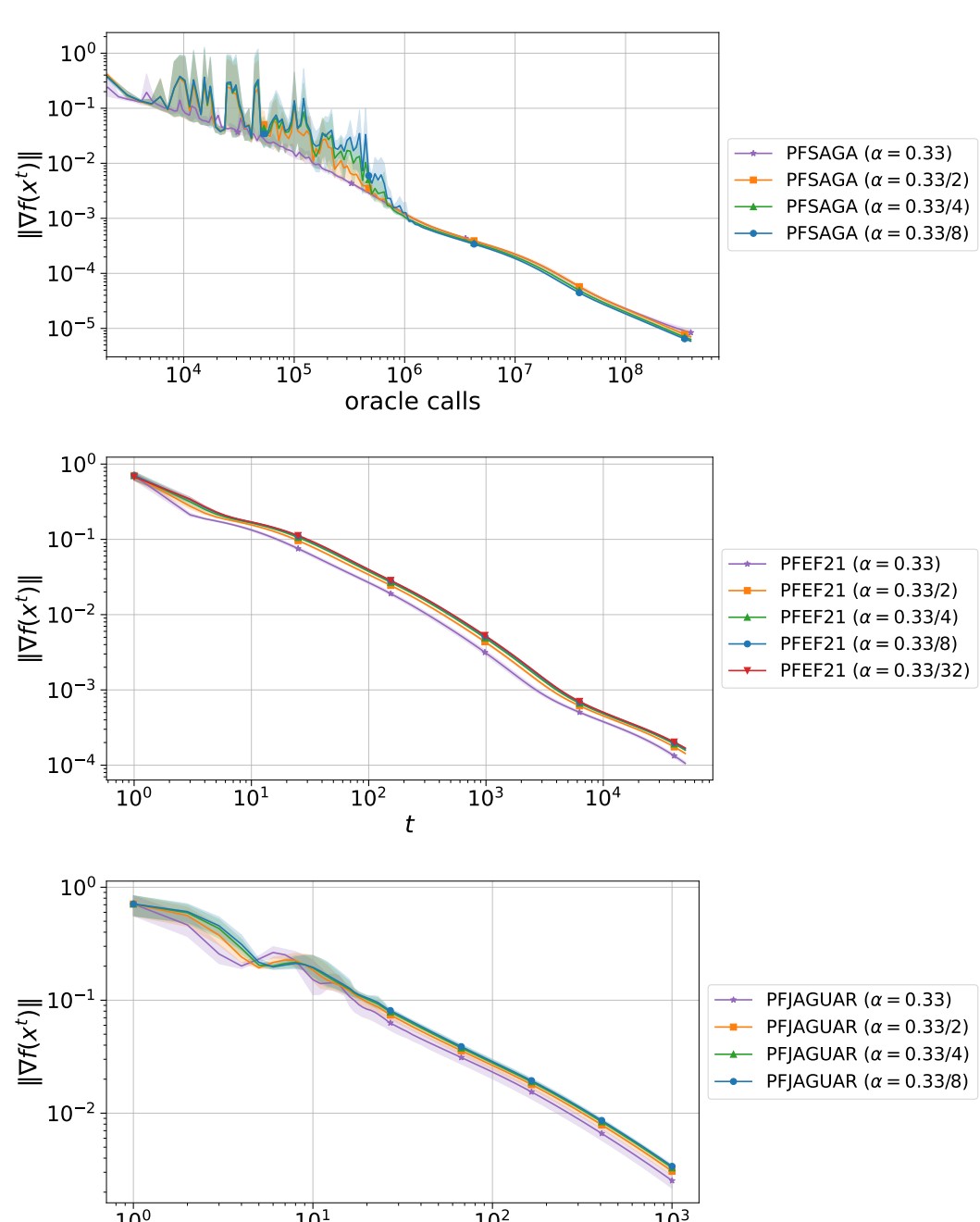

It can be seen, that larger choice of $\alpha$ improves the robustness of the algorithm. However, different $\alpha$ do not influence the overall performance of the algorithm. Justified by this, we take $\alpha = 0.33$ in all experiments afterwards.

## D.2    SAGA ABLATION STUDY

We continue with methods' analyses. We aim to show, that proposed step size scheduler method is valid for different choice of algorithms' hyperparameters, and not only the optimal one. We start with the SAGA method, which depends only on the batch size.

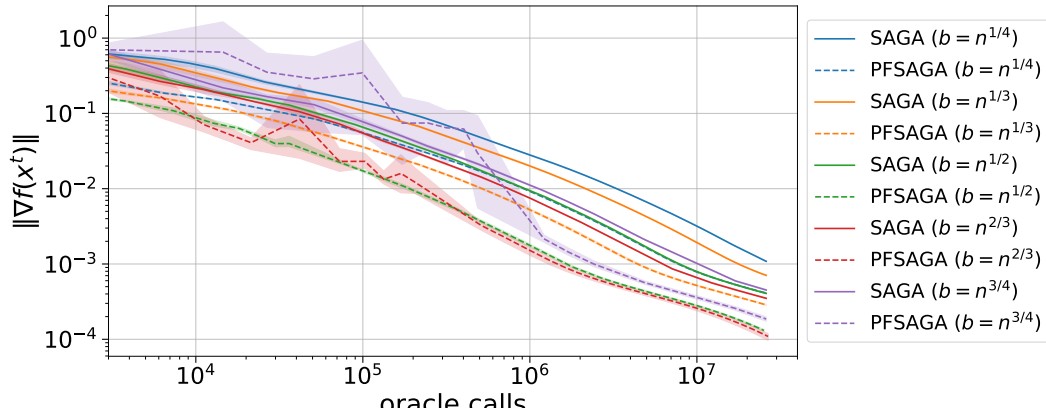

The dotted lines stand for algorithms with adaptive step sizes, while solid lines for method with tuned constant learning rate (8 × theoretical lr). While indeed $n^{2/3}$ being the optimal batch size from both theory and practice, it can be seen, that methods with adaptive stepsize with *any* batch size is better than *any* choice of batch size with constant stepsize.

## D.3    PAGE ABLATION STUDY

We proceed with the PAGE method, whose performance depends on both the batch size and the probability of using a full gradient.

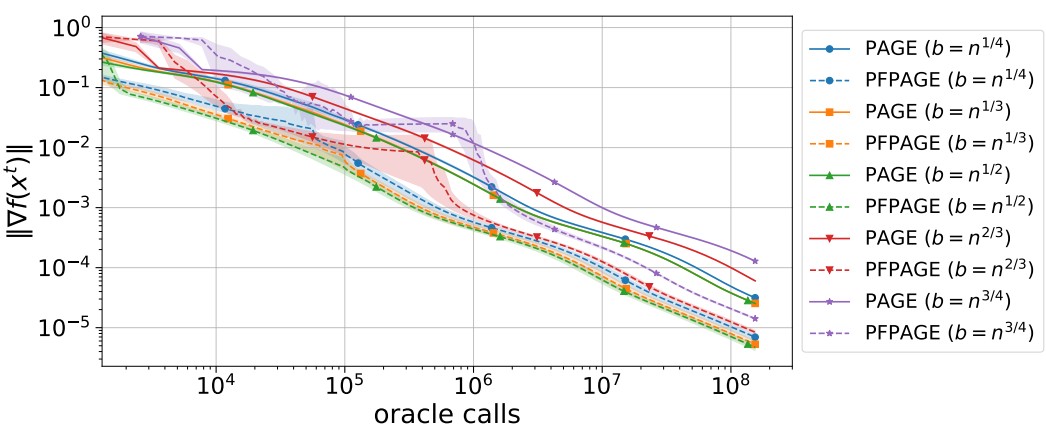

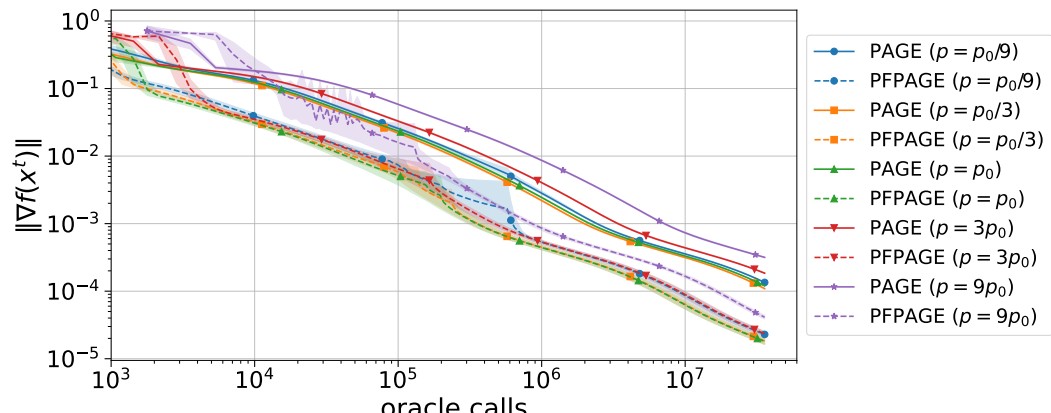

The figure compares adaptive step size scheduling (dotted lines) with the tuned constant step size baseline (solid lines, set to $8\times$ the theoretical value). Ablation Study for batch size $b$ was conducted with the optimal probablity $p$ and vice versa. While tuning both hyperparameters can improve the baseline, adaptive scheduling consistently yields faster convergence across all parameter choices. This indicates that our scheduler reduces the sensitivity of PAGE to its hyperparameters.

### D.4 ZEROSARAH ABLATION STUDY

Next, we consider ZeroSARAH, which involves only the choice of batch size

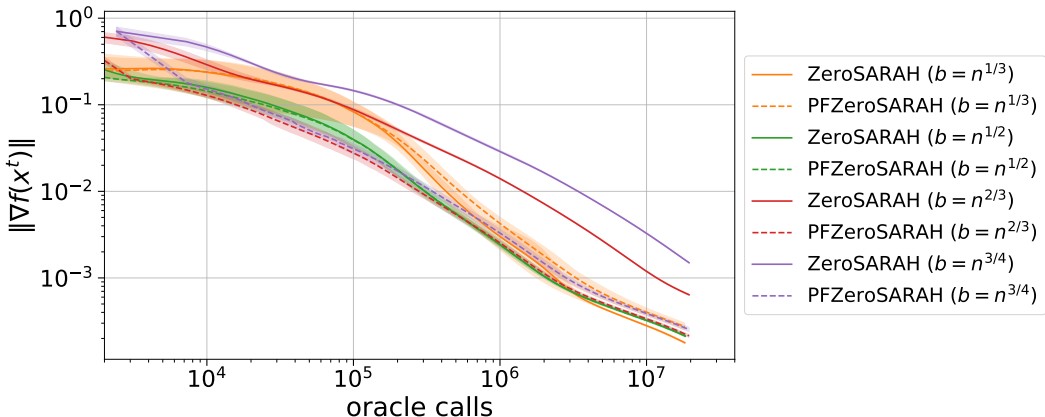

As before, dotted lines represent adaptive step sizes, while solid lines denote tuned constant step sizes ($16\times$ theoretical). The results show that adaptive scheduling makes ZeroSARAH consistently more stable and faster, even when the batch size is not optimally set. Thus, the scheduler effectively compensates for suboptimal hyperparameter choices.

### D.5 EF21 ABLATION STUDY

We now turn to EF21, a method for distributed optimization, based on biased compression with error feedback. Its main hyperparameter is the compression level. We consider Top-k compressor (Alistarh et al., 2018), which preserve $k$ coordinates with maximum absolute value.

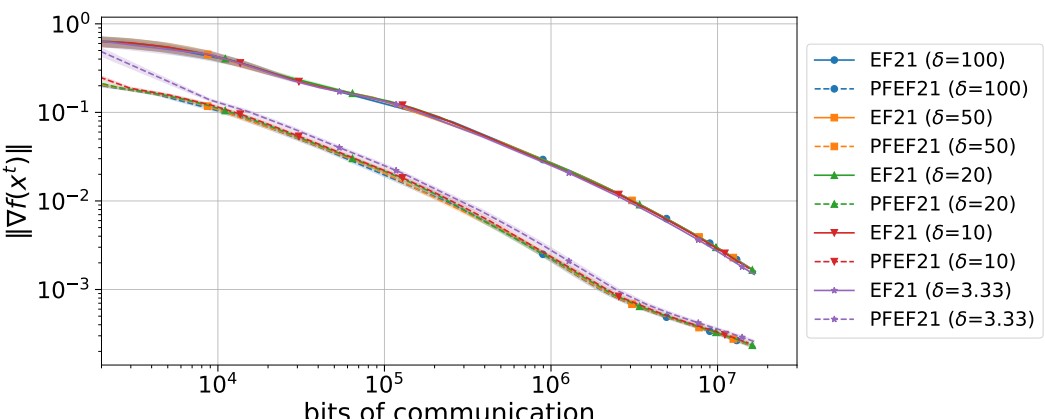

The comparison highlights that adaptive step sizes (dotted) consistently outperform constant step sizes (solid) for all compression levels. The tuned stepsize is $8 \times$ theoretical. Importantly, the advantage persists even when the compression is aggressive, showing that the scheduler mitigates the negative effect of reduced communication accuracy.

### D.6 DASHA ABLATION STUDY

We continue with DASHA, which uses unbiased compression combined with variance reduction. Considered hyperparameters here is the number of local clients and the compression properties. We consider RandK operator, that keeps random $k$ coordinates, while rescaling them to preserve unbiasedness

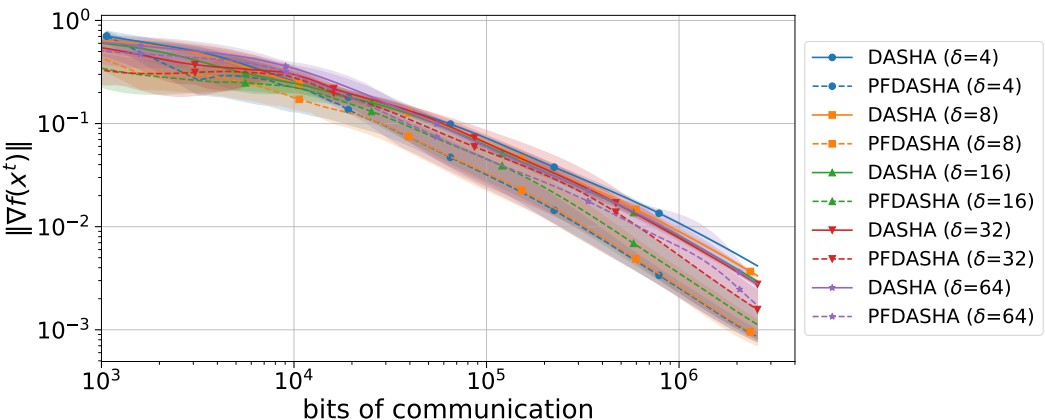

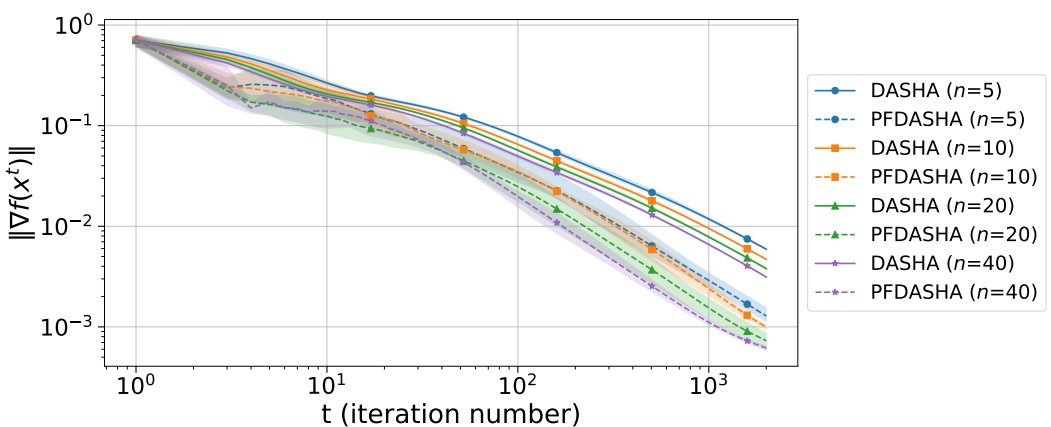

Here too, adaptive step sizes improve convergence speed across different compression ratios. While the constant baseline benefits from careful tuning, it remains inferior to adaptive scheduling in all tested scenarios. This demonstrates that the scheduler provides robustness against the sensitivity of DASHA to compression parameters.

## D.7 JAGUAR ABLATION STUDY

We continue with coordinate-based algorithms. JAGUAR is a method with biased gradient estimators, that did was not included in previous unified frameworks.

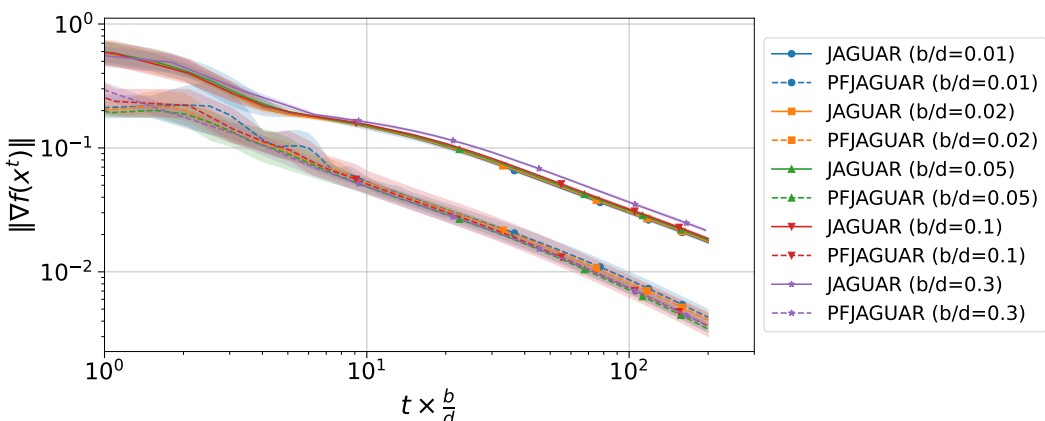

The results indicate that adaptive step sizes maintain a clear advantage across a wide range of update frequencies. Tuned step size is $32 \times$ theoretical. It can be seen, that with wide range of selected number of coordinates, adaptive varaition stays superior to the nonadaptive.

## D.8 SEGA ABLATION STUDY

Finally, we analyze SEGA, which relies on coordinate sketching and depends on the choice of sketch size.

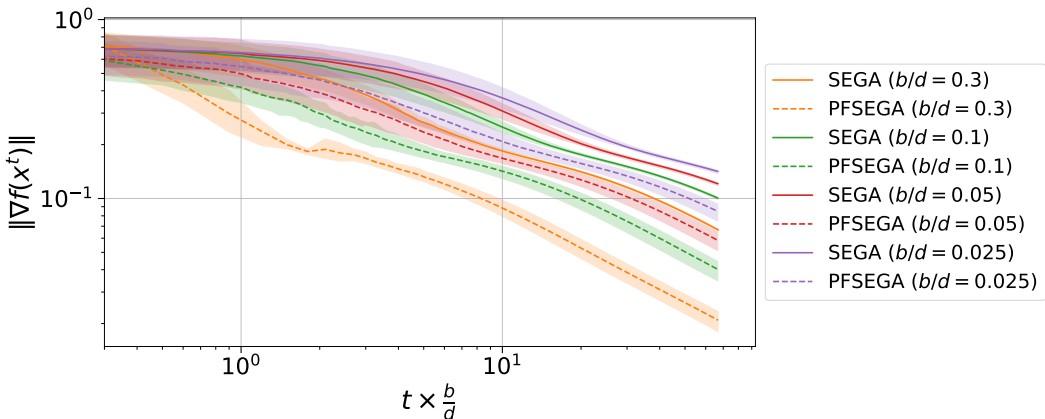

The ablation confirms the same trend: adaptive step sizes (dotted) are better than constant learning rates (solid), regardless of the sketch size. The tuned step size is $32 \times$ theoretical.

These experiments at a9a dataset show, that proposed scheme with adaptive choice of $\gamma_t$ consistently outperforms setups with constant stepsize.

## D.9 STEPSIZE ABLATION STUDY

Further we analyze the behaviour of the adaptive stepsizes throughout the convergence process, compared to the theoretical and tuned constant learning rates.

First of all, we inspect the influence of different $\alpha$ on the step sizes

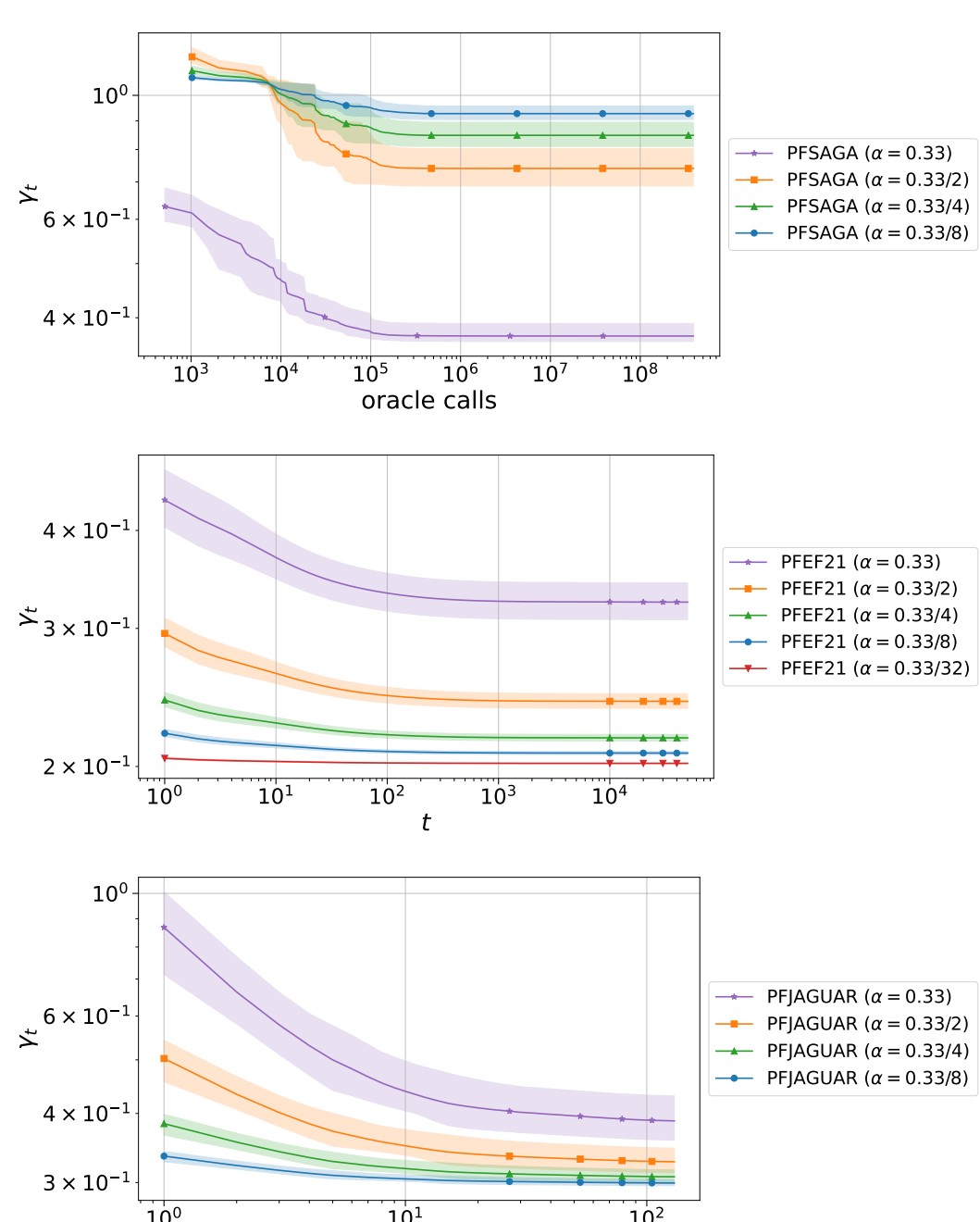

We can notice that stepsizes with $\alpha = 0.33$ differs majorly from others. It can be noted, that learning rate depends monotonically on $\alpha$, however, we cannot tell whether it is increasing, or diminishing.

To validate, that adaptive stepsizes stabilize and are not less, than theoretical, we investigate other methods:

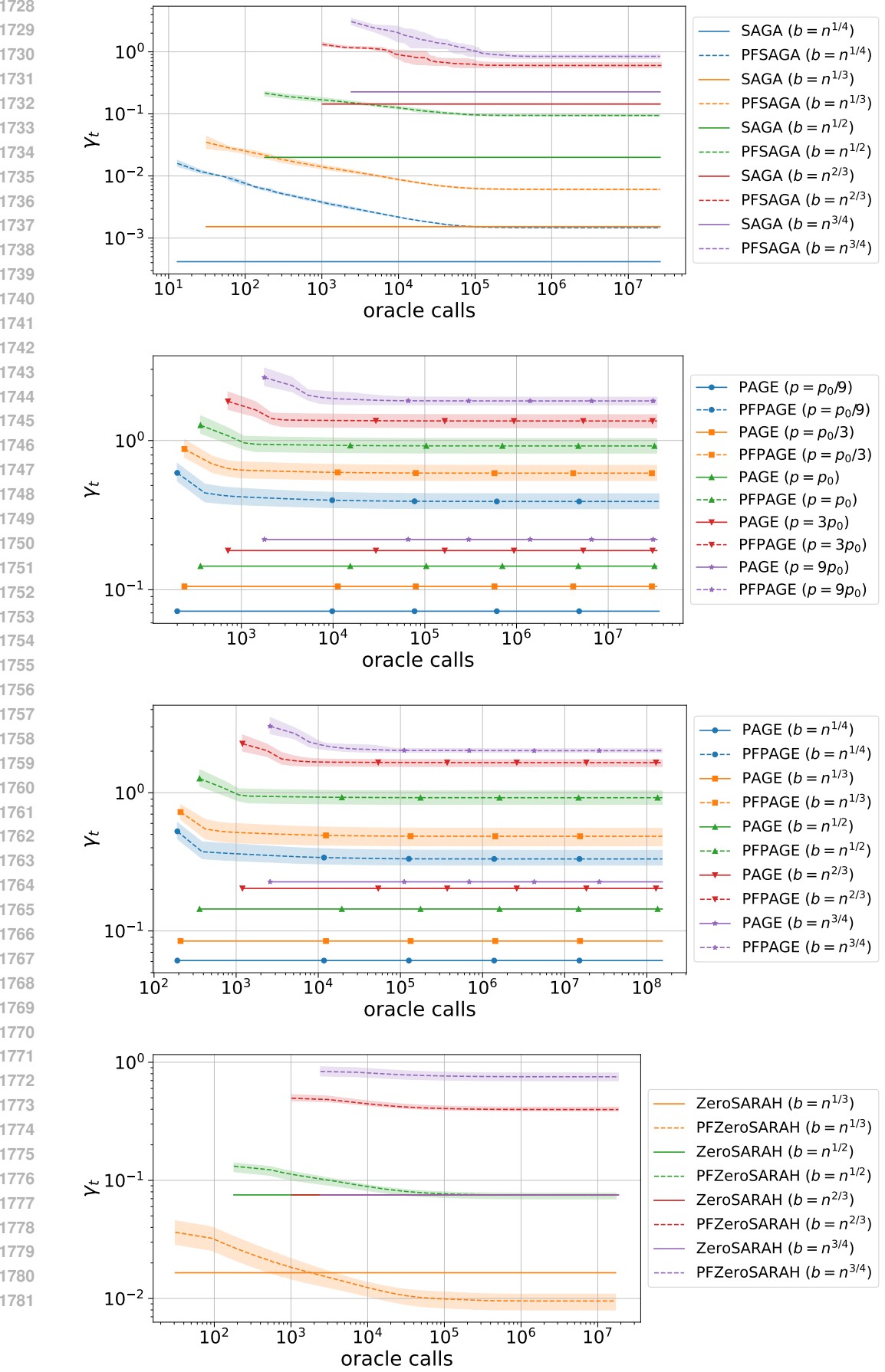

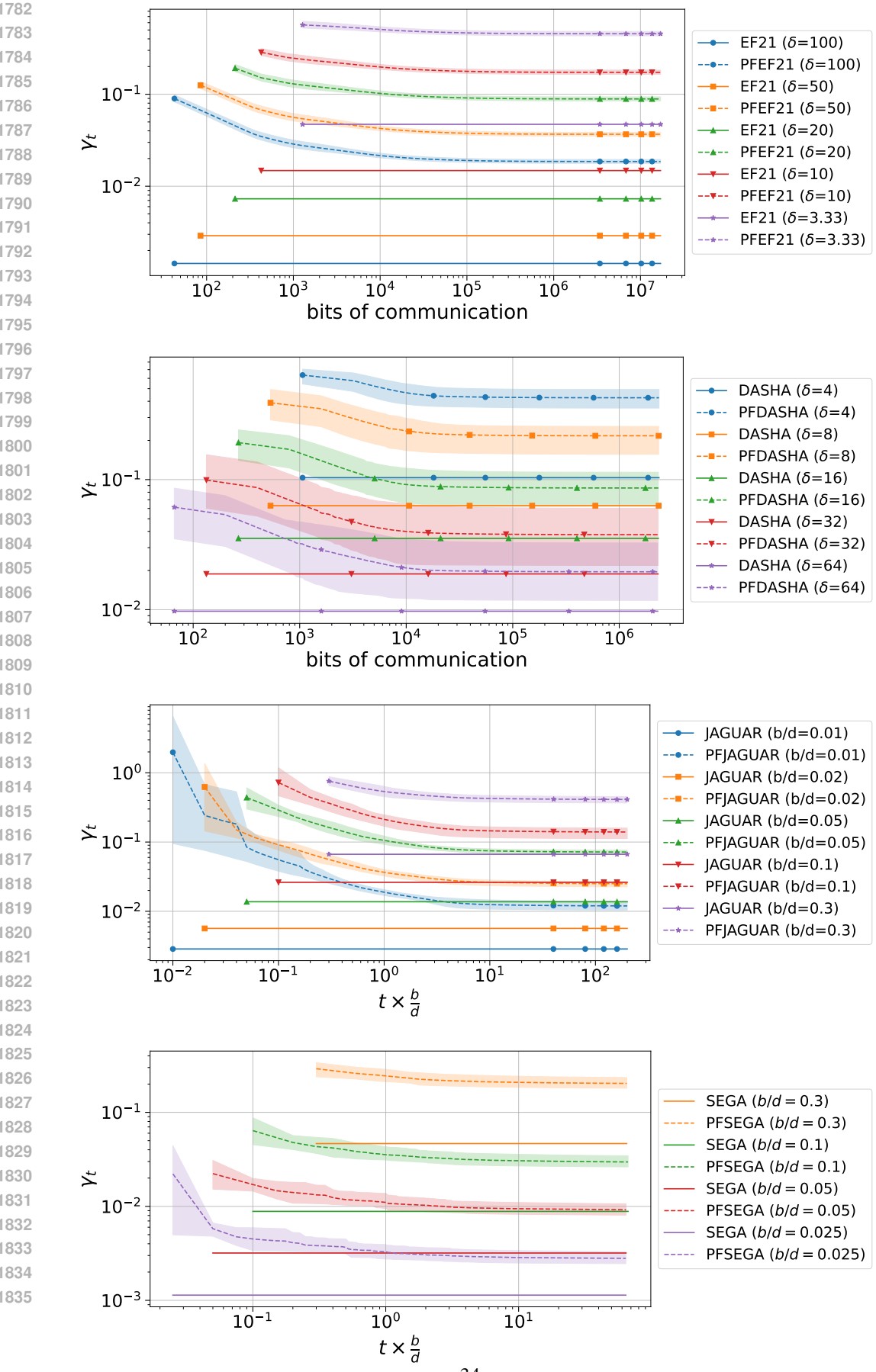

For the tuned step sizes in all the experiments above, we started from the theoretical value and then performed a grid search on a logarithmic grid with base 2. This is reflected in the reported optimal learning rates of the form "$2^n \times$ the theoretical value"

## E    DECLARATION OF LLM USAGE

Large Language Models were not used in the creating process of this manuscript.

