# OpenReview forum: "Unified Theory of Adaptive Variance Reduction"
_ICLR.cc/2026/Conference — Submitted to ICLR 2026_

### Official Review · Reviewer_JVNF · 2025-10-24

**Soundness:** 2
**Presentation:** 1
**Contribution:** 1
**Rating:** 2
**Confidence:** 3

**Summary:**

This paper investigates variance reduction based methods and provides a general theoretical analysis for such algorithms. The authors introduce a unified assumption that does not require the unbiasedness of the stochastic gradient estimator and develop their proofs based on this assumption. They further propose an adaptive variant that eliminates the need for hyperparameter tuning.  Moreover, they extend their analysis to finite-sum problems and distributed optimization. They also conduct numerical experiments to verify the effectiveness of their methods.

**Strengths:**

- This work provides a general theoretical analysis for variance reduction based methods, which may appear interesting.
- The authors give several examples showing that the proposed unified assumption can be satisfied by some existing algorithms.
- The analysis is further extended to other problem settings, although the extensions follow straightforwardly from proof of Section 4.
- The paper also includes numerical experiments to support the theoretical claims.

**Weaknesses:**

- My main concern is about Assumption 1, because after reading the proofs, it seems that the analysis mainly follows from this assumption. By simplifying this assumption, we can get

  $\mathbb{E}[\\| g^{t} - \nabla f(x^{t}) \\|^{2} ]
  < (1 - \rho_{1}) \\| g^{t-1} - \nabla f(x^{t-1}) \\|^{2}+ A \sigma_{t-1}^{2} + B L^{2} \gamma_t^2\\| g^t \\|^{2},$

  which is a more general form of the variance reduction guarantee. For example, the theoretical guarantee of the STORM algorithm is

  $\mathbb{E}[\\| g^{t} - \nabla f(x^{t}) \\|^{2} ]
  < (1 - \beta) \\| g^{t-1} - \nabla f(x^{t-1}) \\|^{2}+ 2\beta^2 \sigma_{t-1}^{2} + 2 L^{2} \gamma_t^2\\| g^t \\|^{2}.$

  Therefore, I am concerned about the contribution and novelty of this work.

- I believe that the overall writing quality of the paper can be improved. For example, I do not agree that “Our Contribution” should appear as a standalone section and the equation in line 780 exceeds the width.

- Although the authors claim that Theorem 3 is truly parameter-free, it seems that the momentum parameter in the variance reduction update may still require manual tuning. How exactly is this parameter determined in practice?

**Questions:**

- In line 189, the authors write that constants B and C are used to bound the difference with the step size.
   However, I am confused by this statement, because as I mentioned in the Weaknesses, this term also includes $\\|g^t\\|^2$.
- I am also curious whether Assumption 1 still holds under the setting of generalized smoothness [1]. Could the authors clarify whether this assumption can be extended to that case?
- In the distributed optimization, the proposed algorithm is based on EF21. However, EF21 uses contractive compressors only in one direction (from workers to the server).  I wonder whether it is possible to employ contractive compressors bidirectionally as [2].



[1] Convex and Non-convex Optimization under Generalized Smoothness. NeurIPS 2023.

[2] Lower Bounds and Nearly Optimal Algorithms in Distributed Learning with Communication Compression. NeurIPS 2022.

**Details Of Ethics Concerns:**

None.

---

> ### Author Response · Authors · 2025-11-24
>
> Dear JVNF, thank you for your careful consideration and questions. We are committed to a productive discussion and address your questions below. Starting with the Weaknesses section:
>
> - Indeed, STORM gradient estimator satisfied the unified Assumption, presented in our paper. Therefore, if it can be incorporated with our adaptive stepsizes and under weaker assumptions ($L$-smoothness, instead of boudned gradients), resulting in optimal $\mathcal{O}\left(\frac{1}{\sqrt{T}}\right)$ convergence. The key point of our paper, is the *equivalence* of all (cooridnate) variance reduction and error compensation methods.
>
> - The quality, transparency, and clarity of the presentation are important to us. We have corrected the formula on line 780 and would appreciate further pointers to any shortcomings. In our view, the "Our Contribution" section is standard and quite useful for understanding the work.
>
> - What exactly do you mean by "momentum parameter in the variance reduction update" in Theorem 3? If you are referring to the parameter $\alpha$, we explicitly state that the analysis guarantees convergence for any value of $\alpha \in (0, \frac{1}{3})$. Furthermore, the explicit form of the bound in Theorem 3 shows that choosing an $\alpha$ close to 0 sharply worsens the estimate. The experiments presented in the paper showed the method's insensitivity to this constant on the chosen benchmark. We thank you for this observation and agree that this discussion should be added to the text to further strengthen the practical applicability of our results.
>
> 1.  Indeed, according to the explicit form of the step in Algorithm 1, $x^t - x^{t-1}$ can be expressed in terms of $g^{t-1}$. Which of these forms is more natural and better reflects the physics of the process is a debatable point; in any case, we do not see this confusing. Please correct us if we are missing the problem.
>
> 2. These are interesting and relevant questions. Thank you for them. In particular, extending the analysis to the generalized smoothness assumption would further enhance the methods' applicability to real-world problems and the contribution to the fundamental understanding of the field, representing a good direction for future research. However, this concept is, in a certain sense, orthogonal to the main focus of this work, and we fear that overloading the presentation with them would negatively impact the quality of perception.
> Definetely, extending the analysis to generalized smoothness is the future research, that we plan to.
>
> 3. As shown in [1], EF21 with bidirectional compressions is also analyzed with unified analysis, mentioned in our related work (see Appendix). All error compensation approaches are analyzed with similar techniques, and, as our approach covers the previous unified analyses, EF21 with bidirectional compressions also can be utilized with adaptive step sizes.
>
> We are committed to improving the quality of our work and would be grateful for further questions and comments. We believe in the contribution of our research and hope for a reassessment of the score.
>
> [1] Gruntkowska, Kaja, Alexander Tyurin, and Peter Richtárik. "EF21-P and friends: Improved theoretical communication complexity for distributed optimization with bidirectional compression." International conference on machine learning. PMLR, 2023.

---

### Official Review · Reviewer_BEP1 · 2025-10-27

**Soundness:** 3
**Presentation:** 2
**Contribution:** 2
**Rating:** 2
**Confidence:** 4

**Summary:**

This paper presents a unified theoretical framework for adaptive variance reduction techniques in stochastic optimization, eliminating the need for hyperparameter tuning related to problem constants like smoothness parameter L. The framework applies to finite-sum problems (e.g., L-SVRG, SAGA, PAGE, ZeroSARAH), distributed optimization, and coordinate methods, achieving optimal rates for non-convex settings. Numerical experiments on logistic regression using the a9a dataset demonstrate the effectiveness of the proposed methods.

**Strengths:**

1. The proposed Assumption 1 captures the recursive behavior of variance-reduced estimators across diverse settings, allowing inclusion of a wider range of methods.
2. The adaptive step sizes, based on accumulated gradient norms, avoid dependence on unknown constants, making them practical for real-world applications. Asymptotically optimal rates are preserved, and theorems provide clear bounds.
3. The work unifies finite-sum, distributed (with compression to reduce communication), and coordinate methods, with new adaptive variants for each.

**Weaknesses:**

1. In my view, the paper’s main weakness is that its conclusions are not new. Prior adaptive methods [1,2] have already achieved the optimal convergence rate for finite-sum problems, and this work does not provide new results. More importantly, the proposed proof technique is not very different from earlier papers [1,2]. Although previous work focused on a single variance-reduction method, this paper’s contribution and level of innovation seem more like extending the previously introduced adaptive variance-reduction approach to other similar variance-reduction techniques. If there are aspects of the analysis that truly differ from prior methods, the authors should highlight them prominently.

2. To attain the optimal convergence rate $n^{1/4} T^{-1/2}$, the method still requires knowledge of $n$, which remains a problem-dependent parameter. If the optimal convergence rate could be achieved without knowing $n$, the contribution would be substantially stronger.

3. Running experiments only on small datasets for logistic regression is clearly insufficient. More comprehensive and realistic experiments would undoubtedly strengthen the paper.

4. I might have misunderstood here, but the paper introduces the PL condition and presents a theorem for the non-adaptive case. However, it seems the authors do not provide an adaptive method under the PL condition. If so, what is the purpose of presenting the PL condition in the paper?

---

[1] (NeurIPS 2022) Adaptive stochastic variance reduction for non-convex finite-sum minimization.

[2] (NeurIPS 2024) Adaptive Variance Reduction for Stochastic Optimization under Weaker Assumptions

**Questions:**

See the Weakness part.

---

> ### Author Response · Authors · 2025-11-24
>
> Dear Reviewer BEP1, thank you for your review. We are committed to constructive discussion and are pleased to address your questions.
>
> 1.  Our contribution is broader than just a parameter-free method with an optimal rate. Please refer to the detailed discussion in the official comment. Additionally, we note that work [1] is widely recognized, though it primarily provides a unified perspective on methods for a setup where optimal rates had already been established. We extend the results further to the practically important case of biased oracles. Furthermore, to the best of our knowlage, our unified analysis for the practically relevant case of parameter-free algorithms (immediately covering an extended family of methods) is the first unified framework in this practically relevant setup. This setting introduces additional challenges for theoretical analysis and is often associated with extra assumptions, which we have managed to avoid. Note that in the field of parameter-free algorithms, there are many different constructions and our scheme does not spend the budget on parameter tuning with restarts [2], and the method's step size does not have a significant dependence on hyperparameters, unlike in [3,4], which is also an advantage of our framework. For additional comparison to other unified analyses, please, see the Appendix of the updated manuscript.
>
>     [1] Eduard Gorbunov, Filip Hanzely, and Peter Richtarik. A unified theory of sgd: Variance reduction, sampling, quantization and coordinate descent.
>
>     [2] Yair Carmon, Oliver Hinder. Making SGD Parameter-Free.
>
>     [3] Ashok Cutkosky, Francesco Orabona. Momentum-Based Variance Reduction in Non-Convex SGD.
>
>     [4] Konstantin Mishchenko, Aaron Defazio. Prodigy: An Expeditiously Adaptive Parameter-Free Learner.
>
> 2.  Firstly, we agree that tuning the remaining hyperparameters of variance reduction methods requires knowledge of the dataset size $n$. However, achieving adaptivity to other parameters is an orthogonal research direction. Research on obtaining adaptivity to other parameters is even scarcer, and at the current stage of the field's development, we do not consider the lack of adaptivity to $n$ a competitive disadvantage of the proposed methods.
>
>     More substantially, the lack of adaptivity to $n$ is a drawback of the proposed *methods*, but not of the *unified analysis* itself. In the analysis, the dependence on $n$ enters through the parameters of Assumption 1, which in turn depend on the specific algorithm. Achieving adaptivity to $n$ while remaining within the scope of unification would require building a separate framework for variance reduction methods, thereby narrowing the class of algorithms considered. Therefore, we view this direction as separate from ours.
>
> 3.  The work includes an analysis of 3 classes of algorithms (variance reduction, federated learning, coordinate-wise). New parameter-free methods are proposed for many algorithms in these classes. Since new methods are being presented, we believe it is necessary first to test them on a classical task under controlled conditions. Moreover, this choice of benchmark (logistic regression on a9a) allowed us to compare several methods from each class with their classical counterparts and some other established representatives. This benchmark also enabled an extensive ablation study for each family of methods, including sweeping over compression levels, the number of coordinates for coordinate-wise methods, batch sizes for variance reduction, the probability $p$ in PAGE, as well as the parameter $\alpha$. We also note that all classical methods required step size tuning, and all experiments were run with multiple random seeds.
>
> 4.  Significant number of existing unified analyses are derived for strongly convex, strongly quasi-convex functions, as well as for functions, satisfying PL condition. To demonstrate our compatibility with previous analyses we also show the covnergence under PL condition for cosntant stepsize. The analysis of adaptive stepsize under PL condition is the future research, that we plan to do.
>
> We believe we have addressed all your questions. We look forward to continuing the discussion. Clarity, transparency, and understandability in presenting our results are important to us. We believe in the contribution of our work and hope for a reassessment of the score in light of the provided clarifications.

---

### Official Review · Reviewer_csP7 · 2025-11-03

**Soundness:** 3
**Presentation:** 3
**Contribution:** 2
**Rating:** 4
**Confidence:** 4

**Summary:**

The paper proposes a unified analysis framework for stochastic variance-reduction methods that drops the traditional unbiasedness requirement on gradient estimators. Within a single assumption capturing recursive error contraction (with constants A, B, C, ρ1, ρ2), the work derives convergence guarantees for finite-sum, distributed, and coordinate methods, and introduces parameter-free, adaptive step-size schedules that do not require smoothness or PL constants. Instantiations include L-SVRG, SAGA, PAGE, ZeroSARAH, EF21, DASHA, SEGA, and JAGUAR. Experiments on logistic regression (a9a) show adaptive variants outperforming theoretical and tuned constant-step baselines and being robust to hyperparameter choices.

**Strengths:**

1. The analysis encompasses biased and unbiased estimators across finite-sum, distributed, and coordinate methods, filling gaps left by prior unified frameworks.
2. The proposed step-size rule depends only on observable quantities, removes reliance on L or µ, and achieves optimal O(1/√T) nonconvex rates and linear PL rates.
3. Ablations over batch sizes, probabilities, compression levels, clients, and coordinate sketch sizes indicate the adaptive variants consistently outperform tuned constant-step baselines.

**Weaknesses:**

1. Evaluation centers on a single dataset (a9a) and a simple logistic-regression task, with no large-scale nonconvex/deep-learning benchmarks, non-iid federated settings, or real distributed systems.
2. Although the schedule is “parameter-free,” users still choose α; constants (A, B, C, ρ) governing bounds are estimator-specific; links between theoretical and deployable code are not fully operationalized.
3. Claims about extending unified analyses to biased estimators and being first to provide adaptive distributed VR need tighter differentiation from recent unified and adaptive frameworks; related work discussion could be broadened.
4. Results are mainly iteration-wise curves without wall-clock time, communication/compute trade-offs, multiple seeds with confidence intervals, or statistical tests; the advantage over “tuned” baselines may depend on tuning effort and settings.

**Questions:**

Please refer to the weaknesses section.

---

> ### Author Response · Authors · 2025-11-24
>
> Dear csP7,
> Thank you very much for your detailed review and for the time you devoted to evaluating our work. We value your perspective and are glad to respond to your comments point by point.
>
> 1. Since we introduce several classes of new algorithms, we believe it is important to begin evaluating their empirical behavior under highly controlled conditions. Moreover, this benchmark choice allowed us to run extensive experiments across all the algorithmic classes we consider, carefully tuning the step sizes for all classical methods and carrying out the extensive ablation study that you identified as a strength.
>
> 2. The constants in Assumption 1 are used only for the theoretical analysis, they are not needed for practical deployment of the algorithms. From a theoretical perspective, our unified analysis can be applied once the constants $A, B, C, \rho$ are specified. Each of these constants has a clear interpretation, which we describe in the paper. The set of methods covered by our framework is quite diverse, and these methods operate in substantially different setups. For example, it is natural that convergence guarantees for federated methods with compression depend on the compression level, while variance reduction methods have guarantees that depend on batch sizes or the probability of computing a full gradient. At the same time, a framework with no input parameters at all cannot deterministically produce different bounds for such distinct settings and therefore cannot capture the specific structure of different problems and methods. In this light, we view Assumption 1 with its constants as a natural and appropriate choice for a unified analysis.
>
>    Concerning the parameter $\alpha$: we explicitly state that our analysis guarantees convergence for any $\alpha \in (0, \tfrac{1}{3})$. This already partially provides a practical guideline: one can choose $\alpha$ within this interval. In addition, from the explicit bound in Theorem 3, one can see that choosing $\alpha$ too close to $0$ undermines the theoretical guarantees by drastically increasing the influence of the $L$ problem parameter on the bound. However, the experiments showed algorithm performance independance on $\alpha$ parameter. We agree that making this discussion more explicit in the paper will facilitate deployment of our algorithm in real-world applications, and we are grateful for this suggestion.
>
> 3. Our work simultaneously relates to adaptive methods and to unified analyses, and the resulting framework covers a broad class of algorithms. This naturally creates many links to different areas within optimization. We aimed to make the Related Work section dense yet focused, providing the necessary background without pulling the narrative too far away from the main line of contributions. We are very keen to make the exposition as comprehensive as possible and would greatly appreciate more concrete pointers to particular works or angles that you feel should be added or emphasized in the related work section.
>
>
> 4. We reported our experimental results in terms of iteration complexity whenever this was sufficient for a direct comparison of the algorithms. In other cases—for example, in the ablation studies over batch size and the probability of computing the full gradient, compression level, and the number of coordinates used per iteration—we presented results in the most relevant units: number of oracle calls, number of transmitted bits, and the iteration index normalized by the number of used coordinates. All experiments were run with multiple random seeds, and the variance is visualized in the plots.
>
>     For the tuned step sizes, we started from the theoretical value and then performed a grid search on a logarithmic grid with base 2. This is reflected in the reported optimal learning rates of the form “$2^n \times$ the theoretical value”, but we agree that the tuning procedure itself should be described more clearly. We have now added this clarification to the paper and thank you for drawing our attention to this omission.
>
> We aim to continue this productive dialogue, believe in the significance of our work, and hope to see its evaluation improved.

---

### Official Review · Reviewer_n7Pi · 2025-11-04

**Soundness:** 2
**Presentation:** 2
**Contribution:** 2
**Rating:** 4
**Confidence:** 4

**Summary:**

This paper proposes a stochastic optimization method by unifying variance reduction methods across diverse settings and introducing parameter-free adaptive steps that eliminate hyperparameter tuning.

**Strengths:**

1. This paper proposes a unified theoretical framework that encompasses both unbiased and biased gradient estimators for variance reduction stochastic methods.

2. A contribution of this paper is the development of adaptive step size schedules that eliminate the need for hyperparameter tuning.

3. It provides comprehensive convergence guarantees for three critical settings including non-convex optimization, PL condition and adaptive step sizes.

**Weaknesses:**

1. While Assumption 1 is the paper's theoretical centerpiece, it provides limited guidance on how to calibrate constants for new VR methods.

2. The adaptive step size relies on a hyperparameter, but the paper provides limited insight into its practical choice or theoretical impact.

3. The related work section (Section 2) mentions adaptive methods like STORM (Cutkosky & Orabona, 2019) and Prodigy (Mishchenko & Defazio, 2023) but fails to provide a direct, quantitative comparison of the proposed adaptive VR methods to these baselines.

4. The experimental results are not convincing. The paper claims applicability to non-convex problems (e.g., neural networks) but only tests convex logistic regression.

5. Several minor typos and notation ambiguities reduce readability and could confuse readers: In Lemma 14 (Section B.1), the statement mentions "Let "—the extra "N" is a typo and should be removed

**Questions:**

1. While Assumption 1 is the paper's theoretical centerpiece, it provides limited guidance on how to calibrate constants for new VR methods.

2. The adaptive step size relies on a hyperparameter, but the paper provides limited insight into its practical choice or theoretical impact.

3. The related work section (Section 2) mentions adaptive methods like STORM (Cutkosky & Orabona, 2019) and Prodigy (Mishchenko & Defazio, 2023) but fails to provide a direct, quantitative comparison of the proposed adaptive VR methods to these baselines.

4. The experimental results are not convincing. The paper claims applicability to non-convex problems (e.g., neural networks) but only tests convex logistic regression.

5. Several minor typos and notation ambiguities reduce readability and could confuse readers: In Lemma 14 (Section B.1), the statement mentions "Let "—the extra "N" is a typo and should be removed

---

> ### Author Response · Authors · 2025-11-24
>
> Dear n7Pi, thank you very much for your comments and for the time you dedicated to reviewing our work. Below, we address your points one by one.
>
> 1. **Constants in Assumption 1.**
> The constants in Assumption 1 are used for the theoretical analysis of the method. They cannot be calibrated for the arbitraty estimator, as the gradient approximation uniquely determines them. Precisely speaking, estimator determines the lower bounds of these constants, but utilizing greater constants only worsens the convergence rates.
>
> 2. **The adaptive parameter $\alpha$ and its influence.**
> Theorem 3 shows the influence of the parameter $\alpha$ on the theoretical guarantees:
> $$
> \frac{1}{T}\sum\limits_{t=0}^{T-1}\mathbb E\left\|\nabla f(x^t)\right\|  \leq\mathcal{O}\left(\frac{V_0^{\frac{1}{2(1-\alpha)}} + L^{\frac{1}{2\alpha}}}{\sqrt{T}}\max\\{\left(\frac{B\rho_2+AC}{\rho_1\rho_2}\right)^{1/4};1\\}\right).
> $$
> We explicitly state that the method converges for $\alpha \in (0, \tfrac{1}{3})$. We also specify the optimal value $\alpha^\star = \arg\min_{\alpha} \left(V_0^{\frac{1}{2(1-\alpha)}} + L^{\frac{1}{2\alpha}}\right)$.
> In a real-world application setting, the quantities $V_0$ and $L$ are not available. However, knowing that convergence is guaranteed for any $\alpha \in (0, \tfrac{1}{3})$ already allows us to choose $\alpha$ from this interval in practice. Moreover, from the explicit form of the bound we can conclude that choosing $\alpha$ too close to $0$ makes the method overly sensitive to problem constant $L$. The experiments presented in the paper show that, on the chosen benchmark, the method is in practice insensitive to this parameter.
>
> 3. **Comparison to STORM and Prodigy.**
> In the paper, we explicitly state that we match state-of-the-art convergence rates (when compared to non–parameter-free versions) and in some cases even obtain optimal convergence rates. In our view, this is the most informative way to assess the resulting methods.
>
>     In the case of stochastic optimization with variance reduction, we explicitly state that *“all method’s adaptive variations obtain the same asymptotic $\mathcal{O}(1/\sqrt{T})$ as non-adaptive. … this convergence rate is optimal in nonconvex setup, therefore, cannot be improved.”*
>
>     Still, if we compare directly to these works, it is important to note that STORM (Cutkosky & Orabona, 2019) also achieves an optimal rate. However, its step size depends on two constants, and the authors state that, for their experiments, they tuned these constants on a simpler task. Moreover, the analysis is carried out under more restrictive assumptions, such as bounded gradient norm. In contrast, our work relies only on standard assumptions (typical for non–parameter-free settings), namely the PL condition and $L$-smoothness.
>
>     Regarding Prodigy (Mishchenko & Defazio, 2023), the authors likewise show optimality of their rate, but the step size depends on additional constants, and their analysis is also conducted under stronger assumptions (bounded gradient norm).
>
> 4. **Experimental design and choice of benchmark.** The paper includes an analysis of three classes of algorithms (variance reduction, federated learning, and coordinate methods), for algorithms in these classes we propose new parameter-free methods. Since these algorithms are conceptually new, we believe it is important to test them on a classical task under controlled conditions. Moreover, this benchmark choice (logistic regression on the a9a dataset) allowed us to compare several methods from each class simultaneously with their classical versions and with other well-known baselines, tuning the step sizes of classical methods optimally. This benchmark also enabled us to perform an extensive ablation study for each class of methods, including sweeps over compression levels, the number of coordinates for coordinate methods, batch sizes for variance reduction, the parameter $\alpha$, and other constants.
>
> 5. **Typos and clarity.**
> We are strongly committed to clear exposition and are grateful for pointing out the issues. We have corrected the Lemma 14, and have cleaned up the notation to improve readability.
>
> We believe we have addressed all of your questions, please let us know if anything remains unclear or unanswered. We are convinced that unified analysis can substantially advance both theoretical understanding and practical progress in optimization, especially in the challenging and practically important setting of parameter-free methods. Therefore, we interested in further constructive dialogue and rating reconsideration.

---

### Author Response · Authors · 2025-11-24

After reviewing the feedback, we recognized the need to concisely restate our contribution.

**Research Landscape:**
-  Unified analyses do exist for non-adaptive algorithms. However, they do not cover the practically important case of biased oracles.
-   There is a field of parameter-free methods.
    -   It lacks a unified analysis.
    -   It does not encompass federated settings.
    -   Theoretical analysis is challenging and often relies on additional restrictive assumptions. Diverse schemes exist, but their practical application can be limited, for instance, by only partially removing dependence on problem parameters and retaining significant sensitivity to hyperparameters that determine the method's step size. It is fair to note that several methods exist which do not suffer from these drawbacks and achieve optimal rates.

**Our Work:**
-   Provides a unified analysis with an expanded scope of applicability to include biased oracles.
-   Transfers this analysis to the parameter-free setting.
-   The analysis in the new setting:
    -   Enables the construction of entire families of parameter-free methods.
    -   Yields, for the first time, distributed and coordinate parameter-free methods.
-   The analysis in both setups does not introduce additional restrictive assumptions.
-   The steps of the proposed parameter-free methods are independent of problem properties. Convergence is proven for an explicit range of the single step hyperparameter, a range which is independent of the problem and even the specific method. Experiments demonstrated the methods' insensitivity to the value of this hyperparameter on the chosen benchmark.

A more detailed exposition of this framework was originally embedded in the work. However, if this was not clear upon reading, please let us know, as we will need to improve this aspect.

We believe that unified analysis is one of the cornerstones of fundamental understanding in the field. Furthermore, it facilitates the development of new methods. We considered extending unification to new, especially practical and challenging settings like biased oracles and parameter-free methods, to be an important milestone for the optimization community. We are committed to a productive discussion, believe in the contribution of our work, and hope for a reassessment of the scores.

---

### Meta-Review · Area_Chair_ZDYt · 2025-12-09

**Summary:**

This paper introduces a unified theoretical framework for variance-reduction methods in stochastic optimization. The approach is able to handle both biased and unbiased estimators, covering finite-sum, distributed, and coordinate-based optimization; its main idea is to design adaptive step-sizes that do not need hyperparameter tuning or unknown quantities such as the smoothness constant.

However, the technical novelties of the paper are also not clear, as reviewers observe the conclusions are similar to existing works and the proofs are somewhat immediate given Assumption 1. Reviewers also express concerns about the scope of the experiments, noting that the current setup mostly focuses on convex logistic regression and do not consider large-scale or non-convex tasks where its advantages should be the largest. Thus, given the abundance of literature on variance reduction techniques, it is not clear that either the theoretical or empirical advances in the paper are enough to warrant acceptance.

**Reviewer Concerns:**

I believe specific technical questions were addressed by the rebuttal, but questions about both the novelties of the algorithm and the scope of the experiments persist, especially given that there does not seem to be a revised version that addresses these concerns.

**Reviewer Scores:**

From the theoretical side, it is not clear that the rebuttals provided by the author(s) would have convinced reviewers about the technical novelties to a sufficient degree. I believe Reviewer JVNF may have needed additional iterations before changing their score, which may not have been possible even with a full discussion period. As the other reviewers all expressed concerns about the experiments and there were no revised document with additional experiments was provided, I do not think the remaining reviewers would have increased their scores.

---

### Decision · Program_Chairs · 2026-01-26

Reject